# DR-SAC: Distributionally Robust Soft Actor-Critic for Reinforcement Learning under Uncertainty

**Mingxuan Cui**[*], **Duo Zhou**[*†]
**Yuxuan Han**[1], **Grani A. Hanasusanto, Qiong Wang, Huan Zhang, Zhengyuan Zhou**[1]
University of Illinois Urbana-Champaign      [1]New York University.

## Abstract

Deep reinforcement learning (RL) has achieved remarkable success, yet its deployment in real-world scenarios is often limited by vulnerability to environmental uncertainties. Distributionally robust RL (DR-RL) algorithms have been proposed to resolve this challenge, but existing approaches are largely restricted to value-based methods in tabular settings. In this work, we introduce Distributionally Robust Soft Actor-Critic (DR-SAC), the first actor–critic based DR-RL algorithm for offline learning in continuous action spaces. DR-SAC maximizes the entropy-regularized rewards against the worst possible transition models within an KL-divergence constrained uncertainty set. We derive the distributionally robust version of the soft policy iteration with a convergence guarantee and incorporate a generative modeling approach to estimate the unknown nominal transition models. Experiment results on five continuous RL tasks demonstrate our algorithm achieves up to $9.8\times$ higher average reward than the SAC baseline under common perturbations. Additionally, DR-SAC significantly improves computing efficiency and applicability to large-scale problems compared with existing DR-RL algorithms. Code is publicly available at https://github.com/Lemutisme/DR-SAC.

## 1 Introduction

The field of deep reinforcement learning (DRL) has witnessed remarkable progress, enabling agents to learn complex behaviors across a wide range of domains, from game playing to robotic control (Arulkumaran et al., 2017; Francois-Lavet et al., 2018; Chen et al., 2024). In particular, offline deep reinforcement learning, which learns policies from fixed datasets without additional environment interaction, has gained increasing attention due to its practical relevance. By eliminating online exploration, this paradigm improves safety guarantees and data efficiency. Among these methods, Soft Actor-Critic (SAC; Haarnoja et al. (2018a;b)) is a principled algorithm based on entropy regularized reinforcement learning, commonly known as the soft value framework. This maximum entropy formulation is grounded in theoretical foundations (Ziebart, 2010) and has been applied to various contexts, including stochastic control (Todorov, 2008; Rawlik et al., 2012; Messaoud et al., 2024) and inverse reinforcement learning (Ziebart et al., 2008; Zhou et al., 2018).

However, a persistent challenge limiting the deployment of offline deep RL in real-world systems is the sensitivity of learned policies to environmental uncertainties (Whittle, 1981; Enders et al., 2024). Policies trained in a nominal environment often exhibit significant performance degradation when deployed under a slightly different one. This model mismatch may arise from uncertain transition and reward functions, observation and actuator noise, parameter variations, or adversarial perturbations.

Distributionally robust reinforcement learning (DR-RL) addresses this challenge by optimizing policies against the worst-case scenario. Instead of assuming a single Markov Decision Process (MDP), DR-RL adopts a Robust Markov Decision Process (RMDP) framework, which includes a set of MDPs defined by an uncertainty set of distributions around the nominal one.

---

[*]Equal Contribution. [†] Corresponding to duozhou2@illinois.edu

Although both value-based (Liu et al., 2022; Lu et al., 2024) and policy-gradient (Wang & Zou, 2022; Kumar et al., 2023) DR-RL algorithms have been proposed, most existing work focuses on performance guarantees and sample complexity in the tabular setting and cannot be deployed in continuous environments. The only notable exception is Robust Fitted Q-Iteration (RFQI; Panaganti et al. (2022)). However, several fundamental research gaps remain. First, RFQI considers uncertainty sets defined by the Total Variation (TV) distance, which is analytically convenient due to its piecewise linear dual formulation, but does not extend to other divergence measures. Second, its non-robust baseline, Fitted Q-Iteration (FQI; Ernst et al. (2005)), is value-based and suffers from critical limitations, including deterministic learned policies, low applicability to high-dimensional action spaces and high sensitivity to errors in the state-action function (Degris et al., 2012). In contrast, actor-critic algorithms combine low-variance value estimation with scalable policy optimization, making them preferred in continuous control benchmarks and practical applications (Konda & Tsitsiklis, 1999; Grondman et al., 2012). Despite their empirical success, no distributionally robust counterpart has been developed. This gap motivates our Distributionally Robust Soft Actor-Critic (DR-SAC), *the first actor–critic-based DR-RL algorithm for offline learning in continuous action spaces*.

In this work, we assume access only to a dataset collected from the training environment, and the transition distribution in the deployment environment lies within an uncertainty set, defined as a Kullback-Leibler (KL) divergence ball centered at the nominal distribution. The goal is to learn a policy that maximizes the soft value function under the worst possible distributions. The main contributions of this work are:

- **Distributionally Robust Maximum Entropy Framework.** We formulate the maximum entropy learning framework under transition uncertainties modeled by KL-divergence-constrained ambiguity sets. Within this framework, we derive distributionally robust soft policy iteration with convergence guarantees and develop distributionally robust counterpart of SAC.

- **Scalable Reformulation via Functional Inner Optimization.** We exploit the interchange property to reformulate the per-$(s, a)$ scalar inner optimization problems into shared functional optimization. This reformulation eliminates the dependence on state–action space dimensionality, enables application to continuous action space and saves training time by over $80.0\%$ compared to the existing DR-RL algorithm RFQI.

- **Generative Modeling for KL-Based Robustness.** We introduce generative models (VAEs) to estimate unknown nominal transition distributions and construct empirical measures while preserving robustness. This design resolves the double-sampling issue inherent in the non-linear KL-divergence dual formulation and enables distributionally robust soft policy learning in offline and continuous control tasks. Extensive experiments across five environments demonstrate that DR-SAC achieves up to $9.8\times$ higher average reward than the SAC baseline under perturbations.

## 1.1 RELATED WORKS

**Robust RL.** The RMDP and Robust Dynamic Programming method were first introduced in (Iyengar, 2005; Nilim & El Ghaoui, 2005) and have been widely studied in planning settings (Xu & Mannor, 2010; Wiesemann et al., 2013; Yu & Xu, 2015). Beyond classical robust formulations, various approaches have been developed to address uncertainty in reinforcement learning, including soft-robustness (Derman et al., 2018; Lobo et al., 2020; Park et al., 2024), risk sensitivity (Tamar et al., 2015; Pan et al., 2019; Singh et al., 2020; Queeney & Benosman, 2023), and adversarial training (Pinto et al., 2017; Zhang et al., 2020; Cheng et al., 2022). In recent years, many distributionally robust RL algorithms have been proposed with provable guarantees in the tabular setting. These include model-free algorithms based on $Q$-learning (Wang et al., 2023; 2024; Liang et al., 2024) and model-based algorithms extending value iteration (Zhou et al., 2021; Panaganti & Kalathil, 2022; Xu et al., 2023; Ma et al., 2023; Liu & Xu, 2024). However, these algorithms are not applicable to continuous action space environments.

**Model-Free Algorithms for Distributionally Robust RL.** In DR-RL, the nominal transition distributions typically appear in the optimization objective but are unknown in practice. To address this challenge, some model-free algorithms assume access to a simulator generating *i.i.d.* samples from the nominal environment (Liu et al., 2022; Zhou et al., 2023; Ramesh et al., 2024), which violates the offline setting. Other algorithms compute empirical transition frequencies from offline datasets (Derman & Mannor, 2020; Clavier et al., 2023; Shi & Chi, 2024), which are not applicable

in continuous spaces. Lastly, Empirical Risk Minimization (ERM) has also been used to estimate the robust objectives under special structures (Mankowitz et al., 2019; Kordabad et al., 2022).

**VAE in Offline RL.** Variational Autoencoders (VAEs) have been widely used in non-robust offline RL. A common application is to estimate the behavior policy from offline data, and impose policy constraints or pessimistic value regularization to mitigate distributional shift (Fujimoto et al., 2019; Wei et al., 2021; Xu et al., 2022; Lyu et al., 2022). VAEs have also been used for state reconstruction and representation learning in RL (Van Hoof et al., 2016). *To the best of our knowledge, this work is the first to incorporate VAE models in a DR-RL algorithm to estimate nominal transition distributions and generate synthetic samples in the absence of a simulator.*

It is worth noting that Smirnova et al. (2019) proposed a algorithm with a similar name. However, the problem formulation is completely different from ours and most DR-RL literature. Their framework accounts for estimation error in the evaluation step and employs KL divergence to constrain deviations from the behavior policy within a single MDP rather than an RMDP.

## 2 FORMULATION

### 2.1 NOTATION AND BASICS OF SOFT ACTOR-CRITIC

A standard framework for reinforcement learning is the discounted Markov Decision Process, formally defined as a tuple $\mathcal{M} = (\mathcal{S}, \mathcal{A}, R, P, \gamma)$, where $\mathcal{S}$ and $\mathcal{A}$ denote the state and action spaces, respectively, both assumed to be continuous in this work. The random reward function modeled as a mapping $R : \mathcal{S} \times \mathcal{A} \mapsto \mathbb{P}([0, R_{\max}])$, where $\mathbb{P}([0, R_{\max}])$ is the set of random variables supported on $[0, R_{\max}]$. The transition distribution is denoted by $P : \mathcal{S} \times \mathcal{A} \mapsto \Delta(\mathcal{S})$, where $\Delta(\mathcal{S})$ is the set of probability measures over $\mathcal{S}$, and $\gamma \in [0, 1)$ is the discount factor. We denote $r = R(s, a)$ as the random reward and $s'$ as the next state drawn from the transition distribution $p_{s,a} = P(\cdot \mid s, a)$. A policy $\pi : \mathcal{S} \mapsto \Delta(\mathcal{A})$ represents the conditional probability over actions given state. We consider the stochastic stationary policy class, denoted by $\Pi$. The entropy of a stochastic policy $\pi$ at state $s$ is defined as $\mathcal{H}(\pi(s)) = \mathbb{E}[-\log \pi(a|s)]$, measuring the randomness of action. The set of integers from 1 to $n$ is denoted as $[n]$.

In maximum entropy RL, the objective comprises the cumulative discounted reward and an entropy regularization term to encourage exploration. Specifically, given an MDP $\mathcal{M}$, the soft value function under policy $\pi$ is defines as

$$V_{\mathcal{M}}^{\pi}(s) = \mathbb{E}\left[\sum_{t=1}^{\infty} \gamma^{t-1}\Big(r_t + \alpha \cdot \mathcal{H}\big(\pi(s_t)\big)\Big) \Bigg| \pi, s_1 = s\right]. \tag{1}$$

The temperature $\alpha \geq 0$ controls the trade-off between reward maximization and policy stochasticity. The optimal soft value and optimal policy are defined as

$$V_{\mathcal{M}}^{\star} = \max_{\pi \in \Pi} V_{\mathcal{M}}^{\pi}, \quad \pi_{\mathcal{M}}^{\star} = \operatorname*{argmax}_{\pi \in \Pi} V_{\mathcal{M}}^{\pi}. \tag{2}$$

Similarly, the soft state-action value function (soft $Q$-function) under policy $\pi$ is defined as

$$Q_{\mathcal{M}}^{\pi}(s, a) = \mathbb{E}\left[r_1 + \sum_{t=2}^{\infty} \gamma^{t-1}\Big(r_t + \alpha \cdot \mathcal{H}\big(\pi(s_t)\big)\Big) \Bigg| \pi, s_1 = s, a_1 = a\right]. \tag{3}$$

For any mapping $Q : \mathcal{S} \times \mathcal{A} \to \mathbb{R}$, Haarnoja et al. (2018a) defined soft Bellman operator as

$$\mathcal{T}^{\pi} Q(s, a) = \mathbb{E}[r] + \gamma \cdot \mathbb{E}_{p_{s,a}, \pi}\left[Q(s', a') - \alpha \log \pi(a' \mid s')\right]. \tag{4}$$

The Soft Actor-Critic (SAC) algorithm updates the policy through soft policy iteration, which is guaranteed to converge in the tabular case. In each iteration, the soft Bellman operator $\mathcal{T}^{\pi}$ is applied to update the estimation of the soft $Q$-function under the current policy $\pi$. The policy is then updated by minimizing the KL divergence between the candidate policy and the exponential of the soft Q-function:

$$\pi_{k+1} = \operatorname*{argmin}_{\pi \in \Pi} D_{\mathrm{KL}}\left(\pi(\cdot \mid s) \,\middle\|\, \exp\left(\frac{1}{\alpha} Q_{\mathcal{M}}^{\pi_k}(s, \cdot)\right) \Big/ Z(s)\right), \ k = 0, 1, \cdots \tag{5}$$

where $D_{\mathrm{KL}}(P \parallel Q) = \mathbb{E}_P\left[\log\left(\frac{P(x)}{Q(x)}\right)\right]$ denotes the KL divergence and the function $Z(\cdot)$ is the normalizing function ensuring that the exponential term defines a valid probability distribution.

## 2.2 ROBUST MARKOV DECISION PROCESS

In real-world RL tasks, the transition distribution $P$ and reward function $R$ in the deployment environment may differ from the environment which the model is trained in or the offline dataset is collected from. Such environmental shifts motivate the study of the Robust Markov Decision Process framework and the goal of learning policies robust to distributional perturbations. Since the analysis and algorithm design for reward perturbations are similar, we assume the reward function $R$ is unchanged and focus only on uncertainty in the transition distributions.

The RMDP framework is denoted as $\mathcal{M}_\delta = (\mathcal{S}, \mathcal{A}, R, \mathcal{P}(\delta), \gamma)$. We consider transition distributions perturbed within a KL-divergence ball. Specifically, let $\mathcal{P}^0 = \{p_{s,a}^0\}_{(s,a)\in\mathcal{S}\times\mathcal{A}}$ be the nominal transition distributions. For each state-action pair $(s,a) \in \mathcal{S} \times \mathcal{A}$ and $\delta > 0$, we define the KL ball centered at $p_{s,a}^0$ as

$$\mathcal{P}_{s,a}(\delta) := \left\{ p_{s,a} \in \Delta(\mathcal{S}) : D_{\mathrm{KL}}(p_{s,a}\|p_{s,a}^0) \leq \delta \right\}. \tag{6}$$

The ambiguity set $\mathcal{P}(\delta)$ is defined as the Cartesian product of $\mathcal{P}_{s,a}(\delta)$ for all pairs $(s,a) \in \mathcal{S} \times \mathcal{A}$, which belongs to the $(s,a)$-rectangular set (Wiesemann et al., 2013).

Under the RMDP framework, the goal is to optimize performance under the worst-case transition model within the ambiguity set. Given $\mathcal{M}_\delta$, the distributionally robust soft value function under policy $\pi$ is defined as

$$V_{\mathcal{M}_\delta}^\pi(s) = \inf_{\mathbf{p}\in\mathcal{P}(\delta)} \mathbb{E}_{\mathbf{p}}\left[ \sum_{t=1}^\infty \gamma^{t-1}\left( r_t + \alpha \cdot \mathcal{H}\big(\pi(s_t)\big) \right) \,\middle|\, \pi, s_1 = s \right]. \tag{7}$$

Similarly, the distributionally robust soft Q-function is given by

$$Q_{\mathcal{M}_\delta}^\pi(s,a) = \inf_{\mathbf{p}\in\mathcal{P}(\delta)} \mathbb{E}_{\mathbf{p}}\left[ r_1 + \sum_{t=2}^\infty \gamma^{t-1}\left( r_t + \alpha \cdot \mathcal{H}\big(\pi(s_t)\big) \right) \,\middle|\, \pi, s_1 = s, a_1 = a \right]. \tag{8}$$

The distributionally robust optimal value and optimal policy are defined as:

$$V_{\mathcal{M}_\delta}^\star(s) = \max_{\pi\in\Pi} V_{\mathcal{M}_\delta}^\pi(s) \quad \text{and} \quad \pi_{\mathcal{M}_\delta}^\star = \operatorname*{argmax}_{\pi\in\Pi} V_{\mathcal{M}_\delta}^\pi(s). \tag{9}$$

## 3 ALGORITHM: DISTRIBUTIONALLY ROBUST SOFT ACTOR-CRITIC

In this section, we develop the Distributionally Robust Soft Actor-Critic algorithm. We first derive the distributionally robust soft policy iteration and establish its convergence to the optimal policy. To improve computational efficiency, we develop a scalable implementation by replacing the per-$(s,a)$ scalar inner optimization with a shared parametric optimization. Lastly, to handle the unknown nominal distribution in offline settings, we incorporate generative modeling to construct the empirical transition measures.

**Assumption 3.1.** To ensure that the policy entropy $\mathcal{H}(\pi(s)) = \mathbb{E}_{a\sim\pi(\cdot|s)}[-\log\pi(a|s)]$ is bounded, we assume $|\mathcal{A}| < \infty$.

*Remark* 3.2. Assumption 3.1 is inherited from the non-robust baseline SAC (Haarnoja et al., 2018a), which establishes theoretical guarantees in the tabular setting while being empirically used in continuous control benchmarks. Our work extends the performance properties of SAC to the DR-RL framework. In Section 3.3, we design a practical algorithm in continuous action spaces.

## 3.1 DISTRIBUTIONALLY ROBUST SOFT POLICY ITERATION

We begin with the distributionally robust soft policy iteration, which iterates between DR soft policy evaluation and DR soft policy improvement. We further show that this iteration is guaranteed to converge to the DR optimal policy.

**DR soft policy evaluation.** For a fixed policy $\pi$, the DR soft $Q$-function is estimated by iteratively applying the distributionally robust soft Bellman operator, which considers the worst possible transition distribution within the uncertainty set. For any bounded mapping $Q : \mathcal{S} \times \mathcal{A} \to \mathbb{R}$, the distributionally robust soft Bellman operator is defined as:

$$\mathcal{T}_\delta^\pi Q(s,a) := \mathbb{E}[r] + \gamma \cdot \inf_{p_{s,a} \in \mathcal{P}_{s,a}(\delta)} \left\{ \mathbb{E}_{p_{s,a},\pi} \left[ Q(s',a') - \alpha \cdot \log \pi(a' \mid s') \right] \right\}. \tag{10}$$

Following Iyengar (2005); Xu & Mannor (2010), the DR soft $Q$-function can be computed via distributionally robust dynamic programming, and $Q_{\mathcal{M}_\delta}^\pi$ is a fixed point of $\mathcal{T}_\delta^\pi$. However, operator $\mathcal{T}_\delta^\pi$ is generally intractable because it requires solving an infinite-dimensional optimization problem over the transition distributions. To address this, we apply the strong duality for worst-case expectations over a KL-divergence ball and derive a equivalent dual formulation.

**Proposition 3.3** (Dual Formulation of the Distributionally Robust Soft Bellman Operator). *Suppose $Q(s,a)$ is bounded, the distributionally robust soft Bellman operator in Equation* (10) *can be reformulated into:*

$$\mathcal{T}_\delta^\pi Q(s,a) = \mathbb{E}[r] + \gamma \cdot \sup_{\beta \geq 0} \left\{ -\beta \log \left( \mathbb{E}_{p_{s,a}^0} \left[ \exp \left( -\frac{V(s')}{\beta} \right) \right] \right) - \beta\delta \right\}, \tag{11}$$

*where*

$$V(s) = \mathbb{E}_{a \sim \pi} \left[ Q(s,a) - \alpha \cdot \log \pi(a \mid s) \right]. \tag{12}$$

Derivation is provided in Appendix B.1. Importantly, the dual form depends only on the nominal transition distribution $\mathcal{P}_{s,a}^0$, instead of an infinite number of distributions in the uncertainty set $\mathcal{P}(\delta)$. Also, the inner optimization problem is reduced to a one-dimensional problem over the scalar $\beta$, rather than infinite-dimensional distributions. Using this tractable dual form operator, the DR soft $Q$-value for a fixed policy $\pi$ can be computed by iteratively applying $\mathcal{T}_\delta^\pi$.

**Proposition 3.4** (Distributionally Robust Soft Policy Evaluation). *For any fixed policy $\pi \in \Pi$, starting from any bounded mapping $Q^0 : \mathcal{S} \times \mathcal{A} \to \mathbb{R}$, define a sequence $\{Q^k\}$ by iteratively applying distributionally robust soft Bellman operator: $Q^{k+1} = \mathcal{T}_\delta^\pi Q^k$. This sequence converges to the DR soft $Q$-value $Q_{\mathcal{M}_\delta}^\pi$ as $k \to \infty$.*

The proof shows that the operator $\mathcal{T}_\delta^\pi$ is a $\gamma$-contraction mapping, with details in Appendix B.2.

**DR soft policy improvement.** The distributionally robust soft policy improvement step is similar to that in SAC, with DR soft $Q$-value $Q_{\mathcal{M}_\delta}$ replacing its non-robust counterpart. The new policy in each update is defined as

$$\pi_{k+1} = \underset{\pi \in \Pi}{\operatorname{argmin}} \, D_{\mathrm{KL}} \left( \pi(\cdot \mid s) \, \middle\| \, \exp \left( \frac{1}{\alpha} Q_{\mathcal{M}_\delta}^{\pi_k}(s, \cdot) \right) \middle/ Z^{\pi_k}(s) \right), k = 0, 1, \cdots \tag{13}$$

With the above policy updating rule, $\pi_k$ has a non-decreasing value with respect to the DR soft $Q$-function. This extends the non-robust soft policy improvement to the uncertain transition distribution case.

**Proposition 3.5** (Distributionally Robust Soft Policy Improvement). *Suppose $|\mathcal{A}| < \infty$, let $\pi_{k+1}$ be the solution to the optimization problem above. Then*

$$Q_{\mathcal{M}_\delta}^{\pi_{k+1}}(s,a) \geq Q_{\mathcal{M}_\delta}^{\pi_k}(s,a), \quad \forall(s,a) \in \mathcal{S} \times \mathcal{A}. \tag{14}$$

The proof is provided in Appendix B.3. The DR soft policy iteration algorithm alternates DR soft policy evaluation and DR soft policy improvement. The following theorem shows convergence to the DR optimal policy, with proof in Appendix B.4.

**Theorem 3.6** (Distributionally Robust Soft Policy Iteration). *Suppose $|\mathcal{A}| < \infty$, starting from any policy $\pi^0 \in \Pi$, the policy sequence $\{\pi^k\}$ converges to the optimal policy $\pi^\star$ as $k \to \infty$.*

**Key Challenges.** Although DR soft policy iteration converges to the optimal policy in the tabular setting, several challenges arise in extending it to continuous action space and offline setting: 1) the DR soft policy evaluation step is computationally expensive at scale due to the per-$(s,a)$ inner optimization over $\beta$; 2) the nominal transition distribution $p_{s,a}^0$ is typically unknown in offline RL tasks, and 3) exact DR soft policy iteration is not directly applicable in continuous action space. We will resolve these issues step by step in the rest of this section.

## 3.2 Solving Dual Optimization using Generative Model

In offline reinforcement learning, the goal is to learn the optimal policy from a pre-collected dataset $\mathcal{D} = \{(s_i, a_i, r_i, s_i')\}_{i=1}^N$, where $(s_i, a_i) \sim \mu$, with $\mu$ denoting the data generation distribution determined by the behavior policy, $r_i = R(s_i, a_i)$ and $s_i' \sim P^0(\cdot \mid s_i, a_i)$. In this section, we address the key computational and modeling challenges of DR soft policy iteration. Specifically, we (i) develop a scalable reformulation of the dual Bellman operator by approximating the per-$(s, a)$ scalar optimization into a shared optimization problem over a function space, and (ii) introduce a generative modeling scheme to estimate unknown nominal transition distributions.

**Dual Reformulation via Functional Optimization.** In DR soft policy evaluation, the Bellman operator $\mathcal{T}_\delta^\pi$ is iteratively applied to the $Q$-function. From the dual form operator in (11), each application requires solving an optimization problem over a scalar $\beta > 0$. While this optimization is tractable, it must be solved separately for every $(s, a)$ pair, which becomes computationally expensive for large-scale problems. To improve training efficiency, we convert a group of scalar optimization problems into a single functional optimization problem. Intuitively, instead of solving for the optimal $\beta^\star$ separately at each state–action pair, we learn a function that approximates these optimal values jointly across the dataset. This can be achieved by the interchange of minimization and integration in decomposable spaces (Rockafellar & Wets, 2009).

Formally, consider the probability space $(\mathcal{S} \times \mathcal{A}, \Sigma(\mathcal{S} \times \mathcal{A}), \mu)$ and let $L^1(\mathcal{S} \times \mathcal{A}, \Sigma(\mathcal{S} \times \mathcal{A}), \mu)$ be the set of absolutely integrable functions on that space, abbreviated as $L^1$. For any $\delta > 0$ and value function $V : \mathcal{S} \to [0, (R_{\max} + \alpha \log|\mathcal{A}|)/(1 - \gamma)]$, define

$$f((s, a), \beta) := -\beta \log \left( \mathbb{E}_{p_{s,a}^0} \left[ \exp \left( -\frac{V(s')}{\beta} \right) \right] \right) - \beta \delta. \tag{15}$$

Assume $|\mathcal{A}| < \infty$, define the admissible function set

$$\mathcal{G} := \left\{ g \in L_1 : g(s, a) \in \left[ 0, \frac{R_{\max} + \alpha \log|\mathcal{A}|}{(1 - \gamma)\delta} \right], \forall (s, a) \in \mathcal{S} \times \mathcal{A} \right\}. \tag{16}$$

**Proposition 3.7** (Interchange of Minimization and Expectation). *For any $\delta > 0$,*

$$\mathbb{E}_{(s,a) \sim \mathcal{D}} \left[ \sup_{\beta \geq 0} f((s, a), \beta) \right] = \sup_{g \in \mathcal{G}} \mathbb{E}_{(s,a) \sim \mathcal{D}} \left[ f((s, a), g(s, a)) \right]. \tag{17}$$

The proof is provided in Appendix B.5. The results allow us to solve a single optimization problem over the function $g$ instead of the $|\mathcal{D}|$ scalar optimization problems. In practice, $g$ is learned jointly across the dataset, substantially reducing training time while preserving robustness.

Based on Proposition 3.7, we define a *functional* DR soft Bellman operator by replacing the scalar $\beta$ with a function $g(s, a)$ and removing the inner optimization. For any function $g \in \mathcal{G}$ and mapping $Q : \mathcal{S} \times \mathcal{A} \to [0, (R_{\max} + \alpha \log|\mathcal{A}|)/(1 - \gamma)]$, let

$$\begin{aligned} \mathcal{T}_{\delta,g}^\pi Q(s, a) :=& \mathbb{E}[r] + \gamma \cdot f((s, a), g(s, a)) \\ =& \mathbb{E}[r] + \gamma \cdot \left\{ -g(s, a) \log \left( \mathbb{E}_{p_{s,a}^0} \left[ \exp \left( -\frac{V(s')}{g(s, a)} \right) \right] \right) - g(s, a)\delta \right\}, \end{aligned} \tag{18}$$

where $V(s) = \mathbb{E}_{a \sim \pi} [Q(s, a) - \alpha \cdot \log \pi(a \mid s)]$.

**Generative Modeling for Nominal Distributions.** In offline RL, we assume that the nominal distributions $\mathcal{P}^0$ are unknown, and no simulator is available to generate additional samples. Under the KL-constrained uncertainty set, the dual optimization problem in the DR soft Bellman operator (both original and functional) is non-linear. Directly estimating the required expectations from the offline dataset $\mathcal{D}$ suffers from the *double-sampling issue*, making empirical risk minimization inapplicable. A detailed discussion is provided in Appendix A.1.

To enable practical implementation of operator $\mathcal{T}_{\delta,g}^\pi$ in continuous space, we incorporate a generative model to estimate the nominal transition distributions. To be specific, we train a variational autoencoder (VAE) model on collected data $(s, a, s') \in \mathcal{D}$ to learn transition $p_{s,a}^0$. The trained VAE

generates next-state samples $\{\tilde{s}'_i\}_{i=1}^m$ and construct an empirical measure $\tilde{p}^0_{s,a}$. For any function $h : \mathcal{S} \mapsto \mathbb{R}$, the empirical expectation is defined as $\mathbb{E}_{s' \sim \tilde{p}^0_{s,a}}[h(s')] = \frac{1}{m} \sum_{i=1}^m h(\tilde{s}'_i)$. We define the *empirical* DR soft Bellman operator as

$$\widetilde{\mathcal{T}}^\pi_{\delta,g} Q(s,a) := \mathbb{E}[r] + \gamma \cdot \widetilde{f}\big((s,a), g(s,a)\big), \tag{19}$$

where

$$\widetilde{f}\big((s,a), \beta\big) = -\beta \log \left( \mathbb{E}_{\tilde{p}^0_{s,a}} \left[ \exp \left( -\frac{V(s')}{\beta} \right) \right] \right) - \beta\delta. \tag{20}$$

## 3.3 Distributionally Robust Soft Actor-Critic

We now extend the action space to the continuous setting and use neural networks to approximate the DR soft value function and policy. We consider RMDP $\mathcal{M}_\delta$ and omit subscripts in $V$ and $Q$. Our algorithm includes a value network $V_\psi(s)$, $Q$-networks $Q_\theta(s,a)$ and a stochastic policy $\pi_\phi(a \mid s)$, parametrized by $\psi, \theta$ and $\phi$. $\bar{\psi}$ and $\bar{\theta}$ are the target network parameters to stabilize training (Mnih et al., 2015). Let $\varphi$ be the parameters of VAE model. We use a parametrized neural network $\mathcal{G}_\eta$ to approximate the function set $\mathcal{G}$.

The core idea of our DR-SAC algorithm is to alternate between *empirical* DR soft policy evaluation and DR soft policy improvement. The loss of $Q$-network is

$$J_Q^{\mathrm{DR}}(\theta) = \mathbb{E}_{(s,a) \sim \mathcal{D}} \left[ \frac{1}{2} \left( Q_\theta(s,a) - \mathcal{T}^\pi_{\delta,\tilde{g}^\star} Q_\theta(s,a) \right)^2 \right], \tag{21}$$

where

$$\tilde{g}^\star = \underset{g \in \mathcal{G}_\eta}{\arg\sup} \, \mathbb{E}_{(s,a) \in \mathcal{D}} \left[ \widetilde{f}((s,a), g(s,a)) \right]. \tag{22}$$

The loss functions of $\psi$, $\phi$ and $\alpha$ are the same as SAC, and the loss function of $\varphi$ is the standard VAE evidence lower bound (ELBO) loss. We adopt the SAC-v1 algorithm (Haarnoja et al., 2018a) with explicit $V$-function, as we observe empirically that including a $V$-network reduces sensitivity to the behavior policy underlying the offline dataset. Ablation studies are presented in Appendix C.3.3. To mitigate overestimation bias, we employ multiple Q-functions $Q_{\theta_i}, (i \in [n])$, train independently, and use the minimum in updating the value and policy networks. This has been shown to outperform the clipped double $Q$-learning ($n = 2$) in offline RL (An et al., 2021). We formally present the Distributionally Robust Soft Actor-Critic in Algorithm 1. Detailed loss functions are provided in Appendix A.2. We also derived a regret bound in Appendix D.

---

**Algorithm 1** Distributionally Robust Soft Actor-Critic (DR-SAC)

---

**Require:** Offline dataset $\mathcal{D} = \{(s_i, a_i, r_i, s'_i)_{i=1}^N\}$, $V$-function network weights $\psi$, $Q$-function network weights $\theta_i, i \in [n]$, policy network weights $\phi$, transition VAE network weights $\varphi$, weight $\tau$ for moving average, function class $\mathcal{G}_\eta$

1: $\bar{\psi} \leftarrow \psi, \bar{\theta}_i \leftarrow \theta_i$ for $i \in [n]$           ▷ Initialize target network weights for soft update
2: **for** each gradient step **do**
3:      $\varphi \leftarrow \varphi - \lambda_\varphi \hat{\nabla}_\varphi J_{\mathrm{VAE}}(\varphi)$              ▷ Update transition VAE weights
4:      Generate samples $\{\tilde{s}'_i\}_{i=1}^m$ from VAE, form empirical measures $\tilde{p}^0_{s,a}$
5:      Compute optimal function $\tilde{g}^\star$ according to (22)
6:      $\psi \leftarrow \psi - \lambda_\psi \hat{\nabla}_\psi J_V(\psi)$                ▷ Update $V$-function weights
7:      $\theta_i \leftarrow \theta_i - \lambda_Q \hat{\nabla}_{\theta_i} J_Q^{\mathrm{DR}}(\theta_i)$ for $i \in [n]$      ▷ Update $Q$-function weights
8:      $\phi \leftarrow \phi - \lambda_\pi \hat{\nabla}_\phi J_\pi(\phi)$                 ▷ Update policy weights
9:      $\alpha \leftarrow \alpha - \lambda_\alpha \hat{\nabla}_\alpha J(\alpha)$                 ▷ Adjust temperature
10:     $\bar{\psi} \leftarrow \tau\psi + (1-\tau)\bar{\psi}, \bar{\theta}_i \leftarrow \tau\theta_i + (1-\tau)\bar{\theta}_i$ for $i \in [n]$    ▷ Update target network weights
11: **end for**
**Ensure:** $\phi$

---

## 4 EXPERIMENTS

The goal of our experiments is to demonstrate the robustness of DR-SAC under environmental uncertainties in offline RL tasks. We measure performance by the average episode rewards under different perturbations, and compare DR-SAC with non-robust baselines and RFQI. To the best of our knowledge, RFQI is the only offline DR-RL algorithm applicable to continuous action spaces. Moreover, extensive ablation studies demonstrate that VAE-based DR-SAC with functional approximation achieves the best trade-off between robustness and computational efficiency.

### 4.1 SETTINGS

We implement SAC and DR-SAC based on the multiple critic version SAC-N (An et al., 2021). Besides RFQI, we also compare DR-SAC with Fitted Q-Iteration (FQI), Deep Deterministic Policy Gradient (DDPG; Lillicrap et al. (2015)), and Conservative Q-Learning (CQL; Kumar et al. (2020)).

We consider *Pendulum*, *Cartpole*, *LunarLander*, *Reacher* and *HalfCheetah* environments from Gymnasium (Towers et al., 2024). For *Cartpole*, we use the continuous action space version in Mehta et al. (2021). For *LunarLander*, we also adopt a continuous action space setting. All algorithms are trained in the nominal environment and evaluated under various perturbations. In our experiments, perturbations include environment parameter changes, random noise added to observed states and random actuator noise applied to actions. Detailed experimental settings are provided in Appendix C.1

### 4.2 PERFORMANCE ANALYSIS

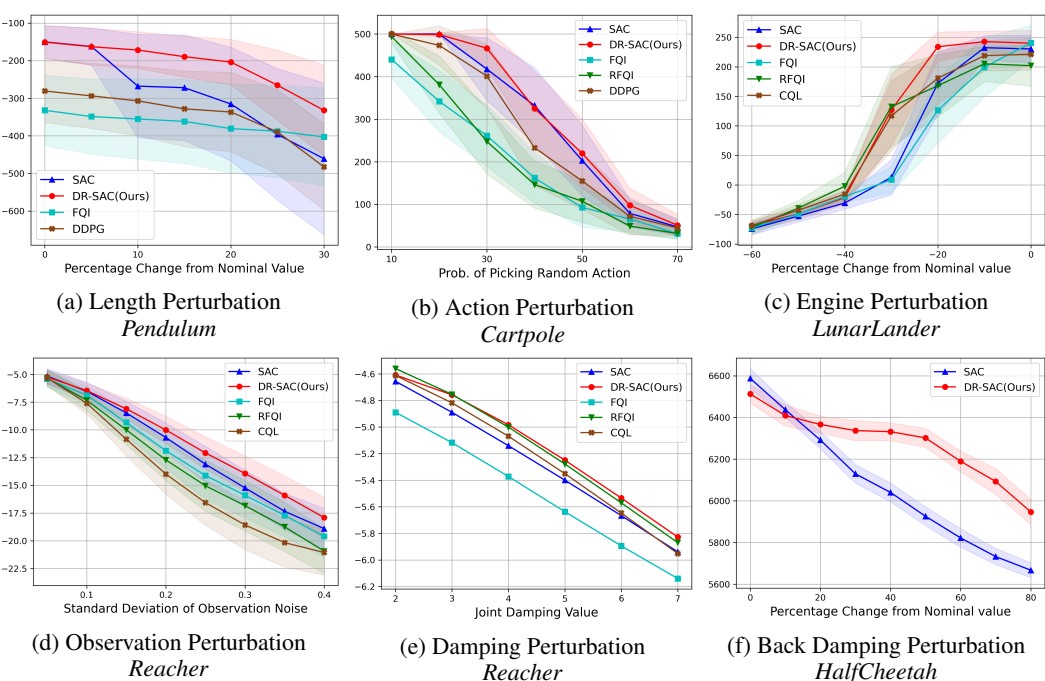

(a) Length Perturbation
*Pendulum*

(b) Action Perturbation
*Cartpole*

(c) Engine Perturbation
*LunarLander*

(d) Observation Perturbation
*Reacher*

(e) Damping Perturbation
*Reacher*

(f) Back Damping Perturbation
*HalfCheetah*

Figure 1: **Robustness performance comparison.** The curves show the average reward over 50 episodes, with shaded regions indicating $\pm 0.5$ standard deviation (see Table 4 for standard deviations in sub-figure (e)). Environmental perturbations include parameter shifts, state and actuator noise.

This section reports selected experiment results. Additional experiments are provided in Appendix C.2. In the *Pendulum* environment, we evaluate robustness against parameter *length* perturbations. RFQI is omitted due to poor performance in the nominal environment. In Figure 1(a), DR-SAC performance outperforms SAC by 35% when the length changes by 20%. In the *Cartpole* environment, the actuator is perturbed by taking random actions with different probabilities. DR-SAC consistently outperforms RFQI, especially when the probability of random action is below 50%. In the *LunarLander* environment, we jointly perturb *main_engine_power* and *side_engine_power* to model engine power disturbance. DR-SAC shows consistently robust performance compared to other algorithms. In

Figure 1(c), under 20% perturbation, DR-SAC achieves an average reward of 240 while the rewards of all other algorithms drop under 180. Moreover, DR-SAC achieves $9.8\times$ higher reward than SAC when parameters reduce by 30%.

To further evaluate robustness in more complex environments, we conduct experiments in *HalfCheetah* and *Reacher* from MuJoCo (Todorov et al., 2012). In the *Reacher* environment, we consider Gaussian observation noise and parameter *joint_damping* perturbation. In Figure 1(d), DR-SAC shows the best performance across all observation noise levels. In Figure 1(e), DR-SAC clearly outperforms SAC. In the *HalfCheetah* environment, we present the experiments of SAC and DR-SAC due to the poor performance of FQI and RFQI. When *back_damping* varies within 50%, DR-SAC maintains a stable average reward of over 6300, while the average reward of SAC keeps decreasing below 5950.

**Discussion on FQI Failure.**    It is worth noting that FQI and RFQI perform poorly even in unperturbed *Pendulum* and *HalfCheetah* environments. One possible reason is that offline RL algorithms exhibit different sensitivities to dataset distributions. SAC works well when the dataset provides a broad coverage over the action space (Kumar et al., 2019). In contrast, FQI is implemented on Batch-Constrained Deep Q-learning (BCQ; Fujimoto et al. (2019)), which restricts the agent to selecting actions close to the behavior policy. This conflicts with the epsilon-greedy-method data generation process in our experiments. One primary goal of our experiments is to demonstrate that DR-SAC improves robustness over SAC under common environmental perturbations. Investigating the sensitivity of offline RL algorithms to dataset distribution is out of the scope of this work.

### 4.3    Ablation Studies

To better understand the design choices in DR-SAC, we conduct a series of ablation studies focusing on computational efficiency and generative model selection. Specifically, we examine (i) the impact of functional approximation on robustness and training time, (ii) the optimization efficiency compared to RFQI, and (iii) the sensitivity of DR-SAC to different generative modeling choices. These studies aim to validate that the proposed design achieves a favorable trade-off between robustness and efficiency.

#### 4.3.1    Training Efficiency of DR-SAC.

**Comparison with Accurate Bellman Operator**    In Section 3.2, we approximate the Bellman operator $\mathcal{T}_\delta^\pi$ with $\mathcal{T}_{\delta,q}^\pi$ to avoid solving optimization problems for each $(s,a)$ pair. To evaluate the impact of this approximation, we additionally implement an algorithm using the accurate operator. We refer to this variant as *DR-SAC-Accurate* and denote Algorithm 1 as *DR-SAC-Functional* in this section. As shown in Figure 2, *DR-SAC-Functional* achieves comparable and even better robustness performance while requiring less than 2% training time. These results validate that functional approximation significantly improves computational efficiency without sacrificing robustness. Variant algorithm details and training time are provided in Appendix C.3.1.

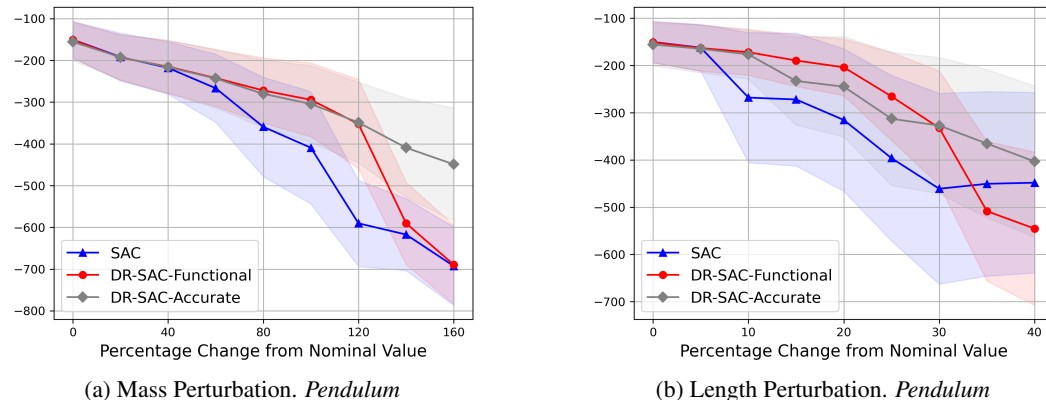

(a) Mass Perturbation. *Pendulum*          (b) Length Perturbation. *Pendulum*

Figure 2: **Efficiency-Robustness Trade-Off.** These figures show that DR-SAC with functional approximation maintains robustness.

**Comparison with RFQI** In Section 4.2, RFQI achieves comparable robustness to DR-SAC in certain environments. However, DR-SAC demonstrates substantial improvement in training efficiency. Table 1 shows that RFQI requires up to $23.2\times$ the training time of DR-SAC. Compared with their non-robust counterparts, RFQI requires at least $11.3\times$ the training time of FQI, while DR-SAC training time is at most $2.6\times$ that of SAC.

Further analysis suggests that this efficiency gap primarily stems from optimization complexity. Although RFQI involves a similar functional approximation step as Equation (22), it requires 1000 gradient descent (GD) steps in each update to find the optimal function. In contrast, DR-SAC requires only 5 GD steps to achieve comparable performance. Empirically, reducing the number of GD steps in RFQI leads to a severe performance drop, even in unperturbed environments (results see Appendix C.3.1). This indicates that the structure of RFQI's loss function inherently results in slower convergence and more optimization steps.

Table 1: Training time in different environments (minute)

| Env | SAC | DR-SAC | FQI | RFQI |
|---|---|---|---|---|
| *Cartpole* | 2 | 4 | 7 | 93 |
| *LunarLander* | 16 | 36 | 17 | 238 |
| *Reacher* | 13 | 32 | 14 | 159 |

### 4.3.2 SELECTION OF GENERATIVE MODEL

Although the VAE models inevitably introduce estimation error when constructing empirical transition measures, we find that DR-SAC is largely insensitive to such modeling choices. As shown in Figure 3(a), varying the VAE latent dimension between 5 and 20 in *Pendulum* does not degrade robustness, and DR-SAC consistently outperforms the SAC baseline.

To demonstrate the choice of VAE over other generative models, we implemented diffusion models and normalizing flows as alternative generative models in DR-SAC. Ablation studies in Figure 3(b) and (c) show that flow-based models exhibit unstable performance, even in the unperturbed *Pendulum* environment. The diffusion-based model achieves comparable robustness but requires at least $4.5\times$ the training time of the VAE-based model. We emphasize that the VAE is not necessarily the optimal choice in all settings. Our DR-SAC algorithm can incorporate alternative transition models depending on the task and computational constraints.

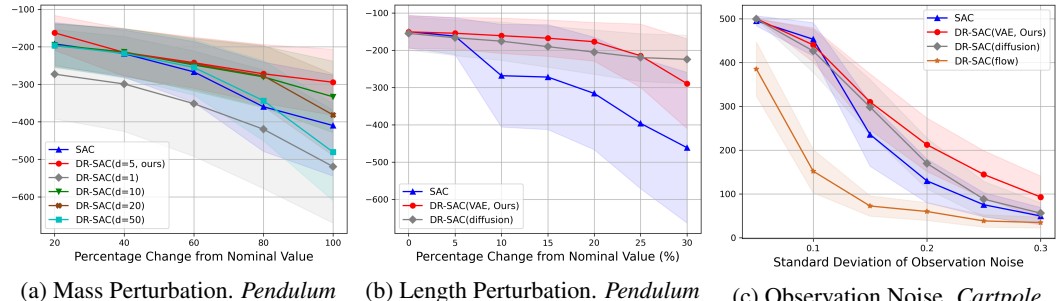

(a) Mass Perturbation. *Pendulum*   (b) Length Perturbation. *Pendulum*   (c) Observation Noise. *Cartpole*

Figure 3: **Generative Model Selection.** Subfigure (a) shows that VAE-based DR-SAC is insensitive to the latent dimension. Subfigures (b) and (c) compare robustness performance with diffusion and flow-based models.

## 5 CONCLUSIONS

We propose DR-SAC, the first actor-critic-based DR-RL algorithm for offline learning in continuous action spaces. Our framework establishes distributionally robust soft policy iteration with convergence guarantees, reduces training time by over 80.0% compared to RFQI through functional optimization, and resolves the double-sampling issue in estimating nominal distributions via generative modeling. Experimental results show that DR-SAC attains up to $9.8\times$ higher reward than SAC under perturbations, highlighting both robustness and efficiency in practical offline RL tasks.

**Acknowledgement.** Grani A. Hanasusanto is supported in part by NSF (CCF2343869 and ECCS-2404413). Huan Zhang is supported in part by the AI2050 program at Schmidt Sciences (AI2050 Early Career Fellowship) and NSF (IIS-2331967). Zhengyuan Zhou is supported in part by NSF (CCF-2312205, ECCS-2419564), ONR-13983263, and the 2027 New York University Center for Global Economy and Business grant. We thank Yunfan Zhang for valuable discussions and the anonymous reviewers for constructive feedback.

**Ethics Statement.** All authors of this submission have read and adhered to the ICLR Code of Ethics.

**Reproducibility Statement.** We provide our code with detailed comments in the supplementary materials. The detailed experiment settings, dataset processing steps and the devices used in our experiments are provided in Appendix C to ensure reproducibility.

**The Use of Large Language Models.** The authors use Large Language Models (LLMs) to assist with grammar checking and language polishing in this submission. LLMs do not play a significant role in research ideation or writing to the extent that they could be regarded as a contributor.

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

# Appendix

CONTENTS

# A   DISCUSSION

## A.1   NECESSITY OF GENERATIVE MODEL

In this section, we explain why model-free empirical risk minimization (ERM) is not applicable under a KL-divergence-constrained uncertainty set and why a generative model (VAE) is necessary.

In offline RL, the nominal transition distributions $\mathcal{P}^0$ are unknown, and no simulator is available to generate additional samples. Under KL-based uncertainty sets, the dual formulation of the DR soft Bellman operator is nonlinear in the transition distribution. As a result, naive empirical estimation from the dataset $\mathcal{D}$ leads to the well-known *double-sampling issue* (Baird et al., 1995), caused by the nested structure of inner and outer expectations. In our algorithm, we want to apply operator $\mathcal{T}_{\delta,g^\star}^\pi$ where $g^\star = \arg\sup_{g \in \mathcal{G}} \mathbb{E}_{(s,a)\sim\mathcal{D}}\left[ f\big((s,a), g(s,a)\big)\right]$. Define the objective function as

$$
\begin{aligned}
J(g) :=& \mathbb{E}_{(s,a)\sim\mathcal{D}}\Big[ f\big((s,a), g(s,a)\big)\Big] \\
=& \mathbb{E}_{(s,a)\sim\mathcal{D}}\left[ -g(s,a) \log\left( \mathbb{E}_{p_{s,a}^0}\left[ \exp\left( \frac{-V(s')}{g(s,a)}\right)\right]\right) - g(s,a)\delta\right].
\end{aligned}
\tag{23}
$$

To construct a consistent empirical estimator of $J(g)$, we would need independent samples to approximate both the outer expectation over $(s,a)$ and the inner expectation over $s' \sim p_{s,a}^0$. This requires splitting the dataset into disjoint subsets $\mathcal{D}_{\text{outer}}$ and $\mathcal{D}_{\text{inner}}$. For each $(s,a) \in \mathcal{D}_{\text{inner}}$, we aggregate the corresponding samples starting from $(s,a)$ contained in $\mathcal{D}_{\text{outer}}$, denoted by $\mathcal{D}_{(s,a)}$, and the empirical risk of objective $J(g)$ becomes

$$
\widehat{J}(g) := \frac{1}{|\mathcal{D}_{\text{out}}|} \sum_{(s,a,s')\in\mathcal{D}_{\text{out}}} \left[ -g(s,a) \log\left( \frac{1}{|\mathcal{D}_{(s,a)}|} \sum_{(\bar{s},\bar{a},\bar{s}')\in\mathcal{D}_{(s,a)}} \exp\left( \frac{-V(\bar{s}')}{g(s,a)}\right)\right) - g(s,a)\delta\right].
\tag{24}
$$

However, in continuous state and action spaces, it is nearly impossible to revisit the exact same state–action pair, leading to empty conditional sample sets $\mathcal{D}_{(s,a)} = \emptyset$ and making ERM infeasible.

Importantly, this difficulty arises from the nonlinear structure of the KL-based dual formulation rather than from our functional optimization approach. Even if we abandon the functional approximation and use the exact dual Bellman operator, the same double-sampling issue persists due to the nested expectation:

$$
\begin{aligned}
\mathcal{L}_Q :=& \mathbb{E}_{(s,a)\in\mathcal{D}} \left[ Q(s,a) - \mathcal{T}_\delta^\pi Q(s,a)\right] \\
=& \mathbb{E}_{(s,a)\in\mathcal{D}} \left[ Q(s,a) - \mathbb{E}[r] - \gamma \cdot \sup_{\beta \geq 0} \left\{ -\beta \log\left( \mathbb{E}_{p_{s,a}^0}\left[ \exp\left( \frac{-V(s')}{\beta}\right)\right]\right) - \beta\delta\right\}\right].
\end{aligned}
\tag{25}
$$

In contrast, under TV-based uncertainty sets, the dual formulation is linear in the transition distribution (Panaganti et al., 2022), which avoids this nested expectation and therefore does not suffer from the same issue.

Existing KL-based distributionally robust RL algorithms overcome this difficulty by using a Monte-Carlo rollouts (Liu et al., 2022; Wang et al., 2023), estimation from transition frequency (Wang et al., 2024), or direct estimation of nominal expectations (Liang et al., 2024). However, these approaches rely on either simulators or repeated visitation of state–action pairs and are therefore not applicable in continuous offline RL settings.

## A.2   ALGORITHM DETAILS

In this section, we present a detailed description of the DR-SAC algorithm. In our algorithm, we use neural networks $V_\psi(s)$, $Q_\theta(s,a)$ and $\pi_\phi(a \mid s)$ to approximate the value function, the $Q$-function and the stochastic policy, respectively, with $\psi, \theta, \phi$ as the network parameters. We also utilize target network $V_{\bar{\psi}}(s)$ and $Q_{\bar{\theta}}(s,a)$, where parameters $\bar{\psi}$ and $\bar{\theta}$ are the exponential moving average of respective network weights. Similar to SAC-v1 algorithm (Haarnoja et al., 2018a), the loss function of $V$-network is

$$
J_V(\psi) = \mathbb{E}_{s\sim\mathcal{D}} \left[ \frac{1}{2} \left( V_\psi(s) - \mathbb{E}_{a\sim\pi_\phi}\left[ Q_{\bar{\theta}}(s,a) - \alpha \log \pi_\phi(a \mid s)\right]\right)^2\right].
\tag{26}
$$

As introduced in Section 3.3, in our algorithm, we modify the loss function of $Q$-network to

$$J_Q^{\mathrm{DR}}(\theta) = \mathbb{E}_{(s,a)\sim\mathcal{D}}\left[\frac{1}{2}\left(Q_\theta(s,a) - \mathcal{T}_{\delta,\widetilde{g}^\star}^\pi Q_\theta(s,a)\right)^2\right],$$

where

$$
\begin{aligned}
\widetilde{g}^\star &= \operatorname*{argsup}_{g\in\mathcal{G}_\eta} \mathbb{E}_{(s,a)\in\mathcal{D}}\left[\widetilde{f}((s,a),g(s,a))\right]\\
&= \operatorname*{argsup}_{g\in\mathcal{G}_\eta} \mathbb{E}_{(s,a)\in\mathcal{D}}\left[-g(s,a)\log\left(\mathbb{E}_{\widetilde{p}_{s,a}^0}\left[\exp\left(\frac{-V_{\bar\psi}(s')}{g(s,a)}\right)\right]\right) - g(s,a)\delta\right].
\end{aligned}
\tag{27}
$$

Optimal dual function $\widetilde{g}^\star$ can be found with backpropagation through $\eta$. We also keep the assumption of policy network in the standard SAC algorithm by reparameterizing the policy using a neural network transformation $a = f_\phi(\epsilon; s)$, where $\epsilon$ is an input noise vector sampled from a spherical Gaussian. The loss of policy is

$$J_\pi(\phi) = \mathbb{E}_{s\sim\mathcal{D},\epsilon\sim\mathcal{N}}\left[\alpha\log\pi_\phi(f_\phi(\epsilon;s)\mid s) - Q_{\bar\theta}(s, f_\phi(\epsilon;s))\right].\tag{28}$$

In the SAC-v2 algorithm (Haarnoja et al., 2018b), the authors propose an automated entropy temperature adjustment method by using an approximate solution to a constrained optimization problem. The loss of temperature is

$$J(\alpha) = \mathbb{E}_{a\sim\pi_\phi}\left[-\alpha\log\pi_\phi(a\mid s) - \alpha\bar{\mathcal{H}}\right],\tag{29}$$

where $\bar{\mathcal{H}}$ is the desired minimum expected entropy and is usually implemented as the dimensionality of the action space.

In addition, we incorporate generative model into our algorithm. VAE is one of the most popular methods to learn complex distributions and has shown superior performance in generating different types of data. In the DR-SAC algorithm, we use VAE to learn the transition function $P^0(s'\mid s,a)$ by modeling the conditional distribution of next states. It assumes a standard normal prior over the latent variable, $p(z) = \mathcal{N}(0, I)$. The encoder maps $(s, a, s')$ to an approximate posterior $q(z\mid s,a,s')$, and the decoder reconstructs $s'$ from the latent sample $z$ and input $(s, a)$. The training loss is the evidence lower bound (ELBO):

$$J_{\mathrm{VAE}}(\varphi) = \mathbb{E}_{q(z|s,a,s')}\left[\|s' - \hat{s}'\|^2\right] + D_{\mathrm{KL}}\left(q(z\mid s,a,s')\,\|\,\mathcal{N}(0,I)\right),\tag{30}$$

where $\hat{s}'$ are the reconstructed states from the decoder.

## B PROOFS

### B.1 PROOF OF PROPOSITION 3.3

We first provide an established result in DRO to compute the worst-case expectation under perturbation in a KL-divergence constrained uncertainty set.

**Lemma B.1** (Hu & Hong (2013), Theorem 1). *Suppose $G(X)$ has a finite moment generating function in the neighborhood of zero. Then for any $\delta > 0$,*

$$\sup_{P:D_{KL}(P\|P_0)\leq\delta} \mathbb{E}_P[G(X)] = \inf_{\beta\geq 0}\left\{\beta\log\left(\mathbb{E}_{P_0}\left[\exp\left(\frac{G(X)}{\beta}\right)\right]\right) + \beta\delta\right\} \tag{31}$$

*Proof of Proposition 3.3.*

$$\mathcal{T}_\pi^\delta Q(s,a) = \mathbb{E}[r] + \gamma \cdot \inf_{p\in\mathcal{P}_{s,a}(\delta)}\left\{\mathbb{E}_{s'\sim p(\cdot|s,a)}\left[\mathbb{E}_{a'\sim\pi(\cdot|s')}[Q(s',a') - \alpha\log\pi(a'|s')]\right]\right\}$$

$$= \mathbb{E}[r] - \gamma \cdot \sup_{p\in\mathcal{P}_{s,a}(\delta)}\left\{\mathbb{E}_{s'\sim p(\cdot|s,a)}[-V(s')]\right\}$$

$$= \mathbb{E}[r] - \gamma \cdot \inf_{\beta\geq 0}\left\{\beta\log\left(\mathbb{E}_{s'\sim p^0(\cdot|s,a)}\left[\exp\left(\frac{-V(s')}{\beta}\right)\right]\right) + \beta\delta\right\} \quad \text{(Lemma B.1)}$$

$$= \mathbb{E}[r] + \gamma \cdot \sup_{\beta\geq 0}\left\{-\beta\log\left(\mathbb{E}_{s'\sim p^0(\cdot|s,a)}\left[\exp\left(\frac{-V(s')}{\beta}\right)\right]\right) - \beta\delta\right\}$$

To apply Lemma B.1, let $P = p(\cdot|s,a)$, $P_0 = p^0(\cdot|s,a)$, and $G(X) = G(s') = -V(s')$. We assume rewards $r = R(s,a)$ are bounded, and the discount factor $\gamma \in [0,1)$. Since $Q(s,a)$ is bounded, we know $V(s')$ is bounded as well. This implies that $G(s') = -V(s')$ has a finite moment generating function (MGF) under the nominal distribution $p^0(\cdot|s,a)$, i.e., $\mathbb{E}_{s'\sim p^0(\cdot|s,a)}[e^{\lambda G(s')}] < \infty$, for $\lambda$ in a neighborhood of zero. This ensures that $G(s')$ has a finite MGF under $P_0$ as required by Lemma B.1. $\qquad\square$

### B.2 PROOF OF PROPOSITION 3.4

Before providing the proof of Proposition 3.4, we present the optimality conditions of Lemma B.1.

**Lemma B.2** (Hu & Hong (2013), Proposition 2). *Let $\beta^\star$ be an optimal solution of the optimization problem in Equation (31). Let $H = \operatorname{esssup}_{X\sim P_0} G(X)$ and $\kappa = \mathbb{P}_{X\sim P_0}(G(X) = H)$. Suppose the assumption in Lemma B.1 still holds, then $\beta^\star = 0$ or $G(X)$ has a finite moment generating function at $1/\beta^\star$. Moreover, $\beta^\star = 0$ if and only if $H < \infty$, $\kappa > 0$ and $\log\kappa + \delta \geq 0$.*

This lemma tells us the optimal solution is unique when $\beta^\star = 0$. This happens if and only if there is a large enough probability mass on the finite essential supremum of $X$, under the distribution center $P_0$. We use this lemma to discuss either $\beta^\star = 0$ or $\beta^\star > 0$ in the following proof.

*Proof of Proposition 3.4.* Similar to the standard convergence proof of policy evaluation, we want to prove that the operator $\mathcal{T}_\delta^\pi$ is a $\gamma$-contraction mapping. Suppose there are two mappings $Q_{1,2} : \mathcal{S}\times\mathcal{A}\to\mathbb{R}$ and define $V_i(s) = \mathbb{E}_{a\sim\pi}[Q_i(s,a)] - \alpha\mathcal{H}(\pi(s))$, $i = 1, 2$. For any state $s\in\mathcal{S}$, we have

$$|V_1(s) - V_2(s)| = |\mathbb{E}_{a\sim\pi}[Q_1(s,a) - Q_2(s,a)]| \leq \|Q_1 - Q_2\|_\infty.$$

Thus, $\|V_1 - V_2\|_\infty \leq \|Q_1 - Q_2\|_\infty$.

Next, for any $\beta > 0$ and $(s,a)$ fixed, define function

$$F_\beta(V) := -\beta\log\mathbb{E}_{p_{s,a}^0}\left[\exp\left(-\frac{V(s')}{\beta}\right)\right] - \beta\delta. \tag{32}$$

Let $\|V_1 - V_2\|_\infty = d$. Then for any $s'\in\mathcal{S}$, $V_2(s') - d \leq V_1(s') \leq V_2(s') + d$. After exponential, expectation, and logarithm operations, monotonicity is preserved. We have

$$-\beta\log\mathbb{E}_{p_{s,a}^0}\left[\exp\left(-\frac{V_2(s')}{\beta}\right)\right] - d \leq -\beta\log\mathbb{E}_{p_{s,a}^0}\left[\exp\left(-\frac{V_1(s')}{\beta}\right)\right]$$

$$\leq -\beta\log\mathbb{E}_{p_{s,a}^0}\left[\exp\left(-\frac{V_2(s')}{\beta}\right)\right] + d.$$

This gives us $|F_\beta(V_1) - F_\beta(V_2)| \leq \|V_1 - V_2\|_\infty$.

Lastly, we reformulate DR soft Bellman operator as

$$\mathcal{T}_\delta^\pi Q(s, a) = \mathbb{E}[r] + \gamma \cdot \sup_{\beta \geq 0} F_\beta(V). \tag{33}$$

Let $\beta_i^\star$ be an optimal solution of $\sup_{\beta \geq 0} F_\beta(V_i)$, $i = 1, 2$. From Lemma B.2, we know $\beta_i^\star$ is unique when $\beta_i^\star = 0$ is optimal. And the optimal value is the essential infimum $H_i$ when $\beta_i^\star = 0$. We want to show $|F_{\beta_1^\star}(V_1) - F_{\beta_2^\star}(V_2)|$ is bounded in all cases of $\beta_i^\star$.

- Case 1: $\beta_1^\star = \beta_2^\star = 0$.

  In this case, the optimal value is the essential infimum value for both $V_i$. We have

  $$|F_{\beta_1^\star}(V_1) - F_{\beta_2^\star}(V_2)| = \left| \operatorname*{essinf}_{s' \sim P_{s,a}^0} V_1(s') - \operatorname*{essinf}_{s' \sim P_{s,a}^0} V_2(s') \right| \leq \|V_1 - V_2\|_\infty.$$

  The last inequality holds because monotonicity is preserved after taking the essential infimum.

- Case 2: $\beta_1^\star = 0$, $\beta_2^\star > 0$, WLOG.

  In this case, we know from optimality that

  $$H_1 = \operatorname*{essinf}_{s' \sim P_{s,a}^0} V_1(s') \geq F_{\beta_2^\star}(V_1), \ H_2 = \operatorname*{essinf}_{s' \sim P_{s,a}^0} V_2(s') \leq F_{\beta_2^\star}(V_2).$$

  Then we have

  $$H_1 - F_{\beta_2^\star}(V_2) \leq H_1 - H_2 \leq \|V_1 - V_2\|_\infty,$$
  $$F_{\beta_2^\star}(V_2) - H_1 \leq F_{\beta_2^\star}(V_2) - F_{\beta_2^\star}(V_1) \leq \|V_1 - V_2\|_\infty.$$

  Thus, $|F_{\beta_1^\star}(V_1) - F_{\beta_2^\star}(V_2)| = |H_1 - F_{\beta_2^\star}(V_2)| \leq \|V_1 - V_2\|_\infty$.

- Case 3: $\beta_1^\star > 0$, $\beta_2^\star > 0$.

  Suppose $F_{\beta_1^\star}(V_1) \leq F_{\beta_2^\star}(V_2)$, WLOG. Then

  $$|F_{\beta_1^\star}(V_1) - F_{\beta_2^\star}(V_2)| = F_{\beta_2^\star}(V_2) - F_{\beta_1^\star}(V_1) \leq F_{\beta_2^\star}(V_2) - F_{\beta_2^\star}(V_1) \leq \|V_1 - V_2\|_\infty,$$

  where the first inequality comes from the optimality of $\beta_1^\star$.

Thus for any $(s, a)$ pair, we have we

$$
\begin{aligned}
|\mathcal{T}_\delta^\pi Q_1(s, a) - \mathcal{T}_\delta^\pi Q_2(s, a)| &= \gamma \cdot \left| \sup_{\beta_1 \geq 0} F_{\beta_1}(V_1) - \sup_{\beta_2 \geq 0} F_{\beta_2}(V_2) \right| \\
&\leq \gamma \cdot \|V_1 - V_2\|_\infty \\
&\leq \gamma \cdot \|Q_1 - Q_2\|_\infty.
\end{aligned}
$$

Since $\mathcal{T}_\delta^\pi$ is a $\gamma$-contraction mapping, the Banach Fixed-Point Theorem implies that the sequence $\{Q^k\}$ convergences to the unique fixed-point of $\mathcal{T}_\delta^\pi$. From Iyengar (2005); Xu & Mannor (2010),this fixed point corresponds to the distributionally robust soft $Q$-value. $\qquad \square$

### B.3 PROOF OF PROPOSITION 3.5

*Proof.* Given $\pi_k \in \Pi$, let $Q_{\mathcal{M}_\delta}^{\pi_k}$ and $V_{\mathcal{M}_\delta}^{\pi_k}$ be the corresponding DR soft $Q$-function and value function. Denote the function for determining the new policy as

$$J_\pi(\pi'(\cdot \mid s)) := D_{\mathrm{KL}}\left( \pi'(\cdot \mid s)) \left\| \exp\left( \frac{1}{\alpha} Q_{\mathcal{M}_\delta}^\pi(s, \cdot) - \log Z^{\pi_k}(s) \right) \right. \right). \tag{34}$$

According to Equation (13), $\pi_{k+1} = \operatorname{argmin}_{\pi' \in \Pi} J_{\pi_k}(\pi')$ and $J_{\pi_k}(\pi_{k+1}) \leq J_{\pi_k}(\pi_k)$. Hence

$$\mathbb{E}_{a \sim \pi_{k+1}}[\alpha \log \pi_{k+1}(a \mid s) - Q_{\mathcal{M}_\delta}^{\pi_k}(s, \cdot) + \alpha \log Z^{\pi_k}(s)]$$
$$\leq \mathbb{E}_{a \sim \pi_k}[\alpha \log \pi_k + (a \mid s) - Q_{\mathcal{M}_\delta}^{\pi_k}(s, \cdot) + \alpha \log Z^{\pi_k}(s)],$$

and after deleting $Z^{\pi_k}(s)$ on both sides, the inequality is reformulated to

$$\mathbb{E}_{a \sim \pi_{k+1}} \left[ Q_{\mathcal{M}_\delta}^{\pi_k}(s, \cdot) - \alpha \log \pi_{k+1}(a \mid s) \right] \geq V_{\mathcal{M}_\delta}^{\pi_k}(s).$$

Next, consider the DR soft Bellman equation:

$$
\begin{aligned}
Q_{\mathcal{M}_\delta}^{\pi_k}(s, a) =& \mathbb{E}[r] + \gamma \cdot \inf_{p_{s,a} \in \mathcal{P}_{s,a}(\delta)} \left\{ \mathbb{E}_{s' \sim p_{s,a}} \left[ V_{\mathcal{M}_\delta}^{\pi_k}(s') \right] \right\} \\
\leq& \mathbb{E}[r] + \gamma \cdot \inf_{p_{s,a} \in \mathcal{P}_{s,a}(\delta)} \left\{ \mathbb{E}_{s' \sim p_{s,a}} \left[ \mathbb{E}_{a' \sim \pi_{k+1}} \left[ Q_{\mathcal{M}_\delta}^{\pi_k}(s', a') - \alpha \log \pi_{k+1}(a' \mid s') \right] \right] \right\} \\
=& \mathcal{T}_\delta^{\pi_{k+1}} \left( Q_{\mathcal{M}_\delta}^{\pi_k} \right)(s, a) \\
&\vdots \\
\leq& Q_{\mathcal{M}_\delta}^{\pi_{k+1}}(s, a), \ \forall (s, a) \in \mathcal{S} \times \mathcal{A}
\end{aligned}
$$

$$(35)$$

where operator $\mathcal{T}_\delta^{\pi_{k+1}}$ is repeatedly applied to $Q_{\mathcal{M}_\delta}^{\pi_k}$ and its convergence is guaranteed by Proposition 3.4. □

### B.4 PROOF OF THEOREM 3.6

*Proof.* By Proposition 3.5, $Q_{\mathcal{M}_\delta}^{\pi_k}$ is non-decreasing with $k$. Since function $Q_{\mathcal{M}_\delta}^{\pi_k}$ is bounded by $(R_{\max} + \alpha \log|\mathcal{A}|)/(1 - \gamma)$, sequence $\{Q_{\mathcal{M}_\delta}^{\pi_k}\}$ converges. Thus policy sequence $\{\pi_k\}$ convergences to some $\pi^\star$. It remains to show that $\pi^\star$ is indeed optimal. According to Equation (13), $J_{\pi^\star}(\pi^\star) \leq J_{\pi^\star}(\pi)$, $\forall \pi \in \Pi$. Using the same argument in proof of Proposition 3.5, we can show that $Q_{\mathcal{M}_\delta}^{\pi}(s, a) \leq Q_{\mathcal{M}_\delta}^{\pi^\star}(s, a)$ for any $\pi \in \Pi$ and $(s, a) \in \mathcal{S} \times \mathcal{A}$. Hence $\pi^\star$ is an optimal policy. □

### B.5 PROOF OF PROPOSITION 3.7

Before providing the proof, we first introduce two technical lemmas. Specifically, Lemma B.4 establishes the *interchange of minimization and integration* property in decomposable spaces. This property has wide applications in replacing point-wise optimality conditions by optimization in a functional space (Shapiro, 2017; Panaganti et al., 2022).

**Lemma B.3** (Rockafellar & Wets (2009), Exercise 14.29). *Function $f : \Omega \times \mathbb{R}^n \mapsto \mathbb{R}$ (finite-valued) is a normal integrand if $f(\omega, x)$ is measurable in $\omega$ for each $x$ and continuous in $x$ for each $\omega$.*

**Lemma B.4** (Rockafellar & Wets (2009), Theorem 14.60, Exercise 14.61). *Let $f : \Omega \times \mathbb{R} \mapsto \mathbb{R}$ (finite-valued) be a normal integrand. Let $\mathcal{M}(\Omega, \mathcal{A}; \mathbb{R})$ be the space of all measurable functions $x : \Omega \to \mathbb{R}$, $\mathcal{M}_f$ be the collection of all $x \in \mathcal{M}(\Omega, \mathcal{A}; \mathbb{R})$ with $\int_{\omega \in \Omega} f(\omega, x(\omega))\mu(d\omega) < \infty$. Then, for any space with $\mathcal{M}_f \subset \mathcal{X} \subset \mathcal{M}(\Omega, \mathcal{A}; \mathbb{R})$, we have*

$$\inf_{x \in \mathcal{X}} \int_{\omega \in \Omega} f(\omega, x(\omega))\mu(d\omega) = \int_{\omega \in \Omega} \left( \inf_{x \in \mathbb{R}} f(\omega, x) \right) \mu(d\omega).$$

*Proof of Proposition 3.7.* First we want to prove $\beta^\star = \arg\sup_{\beta \geq 0} f((s, a), \beta)$ is bounded in interval $\mathcal{I}_\beta := \left[ 0, \frac{R_{\max} + \alpha \log|\mathcal{A}|}{(1 - \gamma)\delta} \right]$ for any $(s, a) \in \mathcal{S} \times \mathcal{A}$. Rewriting the optimization problem to its primal form, it is clear that

$$f((s, a), \beta^\star) = \inf_{p_{s,a} \in \mathcal{P}_{s,a}(\delta)} \mathbb{E}[V(s')] \geq 0.$$

When $\beta$ is greater than $\frac{R_{\max} + \alpha \log|\mathcal{A}|}{(1 - \gamma)\delta}$, it can never be optimal since

$$
\begin{aligned}
f((s, a), \beta) =& - \beta \log \left( \mathbb{E}_{p_{s,a}^0} \left[ \exp \left( -\frac{V(s')}{\beta} \right) \right] \right) - \beta\delta \\
\leq& - \beta \log \left( \exp \left( -\frac{R_{\max} + \alpha \log|\mathcal{A}|}{(1 - \gamma)\beta} \right) \right) - \beta\delta \\
=& \frac{R_{\max} + \alpha \log|\mathcal{A}|}{1 - \gamma} - \beta\delta < 0.
\end{aligned}
$$

Now we know that $f((s,a),\beta)$ is a finite-valued function for each $(s,a) \in \mathcal{S} \times \mathcal{A}$ and $\beta \in \mathcal{I}_\beta$. Also, it is $\Sigma(\mathcal{S} \times \mathcal{A})$-measurable in $(s,a) \in \mathcal{S} \times \mathcal{A}$ for each $\beta \in \mathcal{I}_\beta$ and is continuous in $\beta$ for each $(s,a) \in \mathcal{S} \times \mathcal{A}$. From Lemma B.3, we know that $f((s,a),\beta)$ is a normal integrand.

Moreover, all functions in $\mathcal{G}$ is upper bounded and measurable so $\mathcal{M}_f \subset \mathcal{G} \subset \mathcal{M}((\mathcal{S} \times \mathcal{A}), \Sigma(\mathcal{S} \times \mathcal{A}); \mathbb{R})$. Proposition 3.7 is a direct conclusion of Lemma B.4. $\square$

## C   EXPERIMENT DETAILS

### C.1   MORE SETTING DETAILS

To allow for comparability of results, all tools were evaluated on equal-cost hardware, a Ubuntu 24.04 LTS system with one Intel(R) Core(TM) i7-6850K CPU, one NVIDIA GTX 1080 Ti GPU with 11 GB memory, and 64 GB RAM. All experiments use 12 CPU cores and 1 GPU.

We implement FQI and RFQI algorithms from `https://github.com/zaiyan-x/RFQI`. DDPG and CQL are implemented from the offline RL library *d3rlpy* (Seno & Imai, 2022).

**Hyperparameter Selection**   Across all environments, we use $\gamma = 0.99$ for discount rate, $\tau = 0.005$ for both $V$ and $Q$ critic soft-update, $\alpha = 0.12$ as initial temperature, $|B| = 256$ for mini-batch size, $|\mathcal{D}| = 10^6$ for data buffer size. Actor, Q and V critic and VAE networks are multilayer perceptrons (MLPs) with $[256, 256]$ as hidden dimension. In the *HalfCheetah* and *Reacher* environments, we use two hidden layers in the actor and critic networks. All other networks have one hidden layer.

There are multiple learning rates in our algorithm. Learning rate for VAE network $\lambda_\varphi$ is $5 \times 10^{-5}$ in the *Pendulum* environment and $5 \times 10^{-4}$ in others. In Step 5 of Algorithm 1, optimal function $\widetilde{g}^\star$ is found via backpropagation with learning rate $\lambda_\eta$. All other learning rates $\lambda_\psi$, $\lambda_\theta$, $\lambda_\phi$ and $\lambda_\alpha$ are the same in each environment and represented by $\lambda_\psi$.

Value of learning rates $\lambda_\psi$ and $\lambda_\eta$, number of Q-critics and latent dimensions in VAE are separately tuned in each environment and presented in Table 2.

Table 2: Hyper-parameters selection in SAC and DR-SAC algorithm training.

| Environment | $\lambda_\psi$ | $\lambda_\eta$ | Q-Critic Number | latent dimensions |
|---|---|---|---|---|
| *Pendulum* | $5 \times 10^{-4}$ | $5 \times 10^{-5}$ | 2 | 5 |
| *Cartpole* | $3 \times 10^{-4}$ | $5 \times 10^{-4}$ | 2 | 5 |
| *LunarLander* | $5 \times 10^{-4}$ | $5 \times 10^{-4}$ | 2 | 10 |
| *HalfCheetah* | $3 \times 10^{-4}$ | $5 \times 10^{-5}$ | 5 | 32 |
| *Reacher* | $3 \times 10^{-4}$ | $5 \times 10^{-5}$ | 5 | 10 |

**Offline Dataset**   To ensure a fair performance comparison, all models within each environment are trained on the same offline dataset. Each datasets contains $10^6$ samples, generated by first training a behavior policy and applying the epsilon-greedy method. For most environments, the behavior policy is trained by the Twin Delayed DDPG (TD3; Fujimoto et al. (2018)) implemented from the *d3rlpy* library (Seno & Imai, 2022). In the *Cartpole* environment we use SAC to train the behavior policy. To ensure fair robustness evaluation, all models are trained to achieve the same performance (500, the maximum reward) in the unperturbed *Cartpole* environment. Datasets generated by TD3-trained (or SAC-trained) behavior policies are denoted as TD3-datasets (or SAC-datasets). Additional details, including the algorithm to train behavior policy, training steps and the random-action probability $\epsilon$ are presented in Table 3.

Table 3: Experiment details in dataset generation

| Environment | Behavior Policy Algorithm | Training Steps | Random-Action Probability $\epsilon$ |
|---|---|---|---|
| *Pendulum* | TD3 | $5 \times 10^4$ | 0.5 |
| *Cartpole* | SAC | $5 \times 10^5$ | 0.5 |
| *LunarLander* | TD3 | $3 \times 10^5$ | 0.5 |
| *HalfCheetah* | TD3 | $10^6$ | 0.3 |
| *Reacher* | TD3 | $10^6$ | 0.3 |

### C.2   EXTRA EXPERIMENT RESULTS

***Pendulum***   In the *Pendulum* environment, we compare DR-SAC with SAC, FQI, and DDPG. All models are trained on TD3-dataset. The robust algorithm RFQI does not perform well even in the

nominal environment. To evaluate the robustness of trained models, we change the environment parameters *length*, *mass*, and *gravity*, with nominal values as $1.0$, $1.0$ and $10.0$ respectively. To further show model performance under heavy-tailed perturbation, we also add Cauchy-distributed noise to state observations. The distribution of noise is defined as standard Cauchy distribution multiplied by a parameter *noise scale*. We grind search $\delta \in \{0.1, 0.2, \cdots, 1.0\}$ and find model under $\delta = 0.5$ have the best overall robustness.

DR-SAC shows consistent robustness improvement compared to all other algorithms. The performance under length perturbation is presented in Figure 1(a). In the mass perturbation test, DR-SAC has the best performance in all cases. For example, the average reward is over $40\%$ higher than SAC when mass changes $120\%$. In Figure 4 (b), there is a notable gap between DR-SAC and SAC performance when gravity acceleration changes $40\%$. In Figure 4 (c), DR-SAC achieves consistent the best performance when *noise scale* increases.

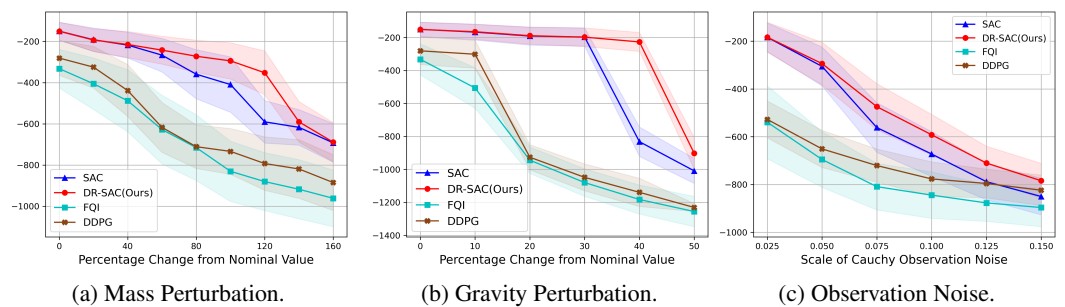

| (a) Mass Perturbation. | (b) Gravity Perturbation. | (c) Observation Noise. |

Figure 4: Additional *Pendulum* experiment results on TD3-dataset.

**Cartpole** In the *Cartpole* environment, we compare the DR-SAC algorithm with non-robust algorithms SAC, DDPG, FQI, and robust algorithm RFQI. All algorithms are trained on the SAC-dataset. In the *Cartpole* environment, the force applied to the cart is continuous and determined by the actuator's action and parameter *force_mag*. The highest possible reward is 500 in each episode. To ensure fair comparison, all models are trained to have average rewards of 500 in the nominal environment.

We test the robustness by introducing two changes to the environment: applying action perturbation and adding observation noise. In the action perturbation test, the actuator takes random actions with different probabilities. In the observation perturbation test, noise with zero mean and different standard deviations is added to the nominal states in each step. We grind search $\delta \in \{0.25, 0.5, 0.75, 1.0\}$ and find DR-SAC has the best performance when $\delta = 0.75$. We also use $\rho = 0.75$ to train the RFQI model.

In the *Cartpole* environment, DR-SAC has the best overall performance under both type of perturbation. Figure 5(a) extends Figure 1(b). DR-SAC has performance improvement over 75% compared to non-robust algorithms SAC and DDPG when the standard deviation of noise is 0.2 and 0.3.

**LunarLander** In the *LunarLander* environment, we compare DR-SAC with non-robust algorithms SAC, CQL, FQI, and robust algorithm RFQI. All algorithms are trained on TD3-dataset. In the *LunarLander* environment, the lander has main and side engines, and the actuator can control the throttle of the main engine. We change environment parameters *engine_power* (main and side engine power) and *wind_power* (magnitude of linear wind) to validate algorithm robustness. We grind search $\delta \in \{0.25, 0.5, 0.75, 1.0\}$ and find DR-SAC has the best performance when $\delta = 0.25$. We also use $\rho = 0.25$ to train the RFQI model.

Under all types of perturbations, DR-SAC shows superior robustness compared to other algorithms. The performance under *engine_power* perturbation is presented in Figure 1(c). In Figure 5(b), DR-SAC shows the highest average reward in most levels of wind perturbation. It is worth noting that the robust algorithm RFQI does not have an acceptable performance in this test, even compared to its non-robust counterpart FQI.

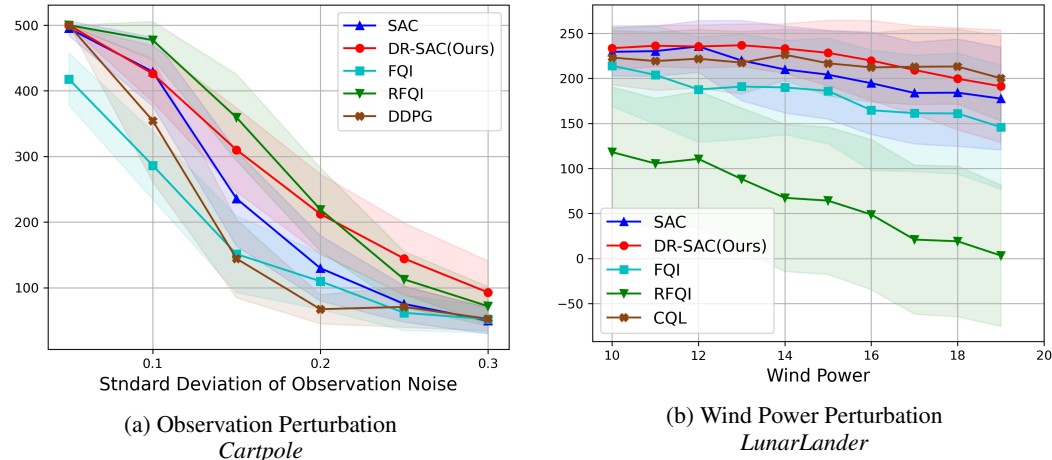

(a) Observation Perturbation
*Cartpole*

(b) Wind Power Perturbation
*LunarLander*

Figure 5: Additional *Cartpole* results on SAC-dataset and *LunarLander* results on TD3-dataset.

**Reacher**   In the *Reacher* environment, we compare DR-SAC with non-robust algorithms SAC, FQI, CQL, and robust algorithm RFQI. All algorithms are trained on TD3-dataset. In *Reacher* environment, the actuator controls a two-jointed robot arm to reach a target. We use *joint_damping* to denote the damping factor of both *joint0* and *joint1*, with default value as $1.0$. We grind search $\delta \in \{0.1, 0.2, 0.3\}$ and find DR-SAC has the best performance when $\delta = 0.2$. We also use $\rho = 0.2$ to train the RFQI model.

To test the robustness of all algorithms, we compare their performance after adding observation noise and changing parameters *joint_damping*. In the observation perturbation test, we add zero-mean Gaussian noise to the nominal state in dimensions $4 - 9$. The first 4 dimensions in state are trigonometric function values and are kept unperturbed. Performance under both perturbations is presented in Figure 1 (d) and (e). Moreover, in Figure 1 (e), the standard deviation regions were computed but omitted from the final plot because the overlapping shaded areas of multiple algorithms made the figure unreadable. We provide them in Table 4.

Table 4: Standard Deviation of Model Performance under Damping Perturbation in *Reacher*

| Joint Damping Value | SAC | DR-SAC | FQI | RFQI | CQL |
|---|---|---|---|---|---|
| 2.0 | 1.452 | 1.459 | 1.638 | 1.509 | 1.396 |
| 3.0 | 1.534 | 1.540 | 1.754 | 1.601 | 1.456 |
| 4.0 | 1.631 | 1.628 | 1.865 | 1.699 | 1.529 |
| 5.0 | 1.735 | 1.721 | 1.969 | 1.802 | 1.614 |
| 6.0 | 1.841 | 1.819 | 2.057 | 1.910 | 1.711 |
| 7.0 | 1.946 | 1.925 | 2.135 | 2.018 | 1.816 |

**HalfCheetah**   In the *HalfCheetah* environment, we compare DR-SAC with SAC baseline only due to the unsatisfactory performance of FQI and RFQI. All algorithms are trained on TD3-dataset. In the *HalfCheetah* environment, the actuator controls a cat-like robot consisting of $9$ body parts and $8$ joints to run. We use *front_stiff* and *front_damping* to denote the stiffness and damping factor of joint *fthigh*, *fshin*, and *ffoot*. Also, *back_stiff* and *back_damping* can be denoted in a similar way. The default value of these parameters can be found through the environmental assets of Gymnasium MuJoCo in `https://github.com/Farama-Foundation/Gymnasium/blob/main/gymnasium/envs/mujoco/assets/half_cheetah.xml`. We grind search $\delta \in \{0.1, 0.2, 0.3\}$ and find DR-SAC has the best performance when $\delta = 0.2$.

Performance of *back_damping* test is presented in Figure 1(f). Combining it with Figure 6, we can see DR-SAC has notable robustness improvement across all perturbation tests. For example, in *front_stiff* perturbation test, DR-SAC achieves an improvement as much as $10\%$ when the change is $80\%$.

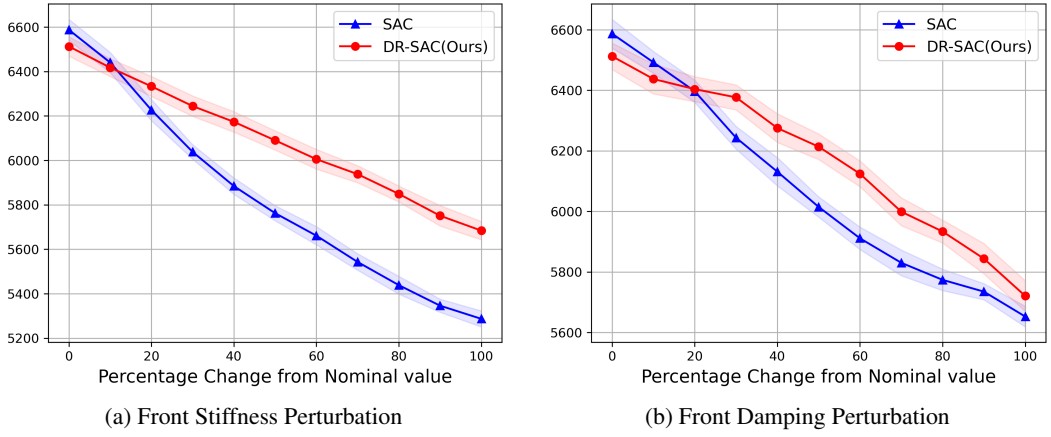

(a) Front Stiffness Perturbation       (b) Front Damping Perturbation

Figure 6: Additional *HalfCheetah* results on TD3-dataset.

## C.3 ABLATION STUDY DETAILS

### C.3.1 TRAINING EFFICIENCY OF DR-SAC.

In this section, we want to show that DR-SAC with functional optimization finds a good balance between efficiency and robustness. We compare training time and robustness of Algorithm 1, DR-SAC without functional optimization, and robust algorithm RFQI, to show our DR-SAC algorithm has the best overall performance.

**Comparison with Accurate Bellman Operator**  We first introduce DR-SAC algorithm without functional optimization. Most steps are the same as Algorithm 1, instead of following modifications. Step 5 in Algorithm 1 is removed. $Q$-network loss is replaced by

$$J_Q^{\text{DR\_acc}} = \mathbb{E}_{(s,a)\sim\mathcal{D}} \left[ Q_\mathcal{M}^\pi(s,a) - \widetilde{\mathcal{T}}_\delta^\pi Q_{\mathcal{M}_\delta}^\pi(s,a) \right]^2, \tag{36}$$

where $\widetilde{\mathcal{T}}_\delta^\pi$ is the empirical version of $\mathcal{T}_\delta^\pi$ by replacing $p_{s,a}^0$ with $\widetilde{p}_{s,a}^0$. We call this modified algorithm *DR-SAC-Accurate* and call Algorithm 1 *DR-SAC-Functional* in this section.

We train SAC, *DR-SAC-Functional*, and *DR-SAC-Accurate* algorithms in *Pendulum* environment. The optimization problem in Equation (11) is a problem over scalar $\beta > 0$ and solved via *Scipy* for each $(s,a)$ pair. Table 5 presents the training steps and time for three algorithms. Training time of *DR-SAC-Accurate* is over $150\times$ longer than standard SAC and over $50\times$ longer than *DR-SAC-Functional*. Considering *Pendulum* environment is relatively simple, *DR-SAC-Accurate* algorithm is hard to utilize in large-scale problems.

Table 5: Training steps and time for three algorithms in *Pendulum*

| Algorithm | Training Steps | Training Time (Minute) |
|---|---|---|
| SAC | 10k | 1.7 |
| *DR-SAC-Functional* | 10k | 4.7 |
| *DR-SAC-Accurate* | 8k | 260 |

Moreover, we test the robustness of three algorithms by comparing their average reward under different perturbations. To be specific, we change *Pendulum* environment parameters: *length* and *mass*. *DR-SAC-Functional* and *DR-SAC-Accurate* are trained with $\delta = 0.5$. Figure 2 shows that *DR-SAC-Functional* achieves comparable and even better performance under small-scale perturbation. For example, *DR-SAC-Functional* and *DR-SAC-Accurate* have almost the same performance under *mass* perturbation test when change is less than $120\%$. In *length* perturbation test, *DR-SAC-Functional* has better performance when the change is less than $30\%$.

**Efficiency Comparison with RFQI.**  In Section 4.2, existing DR-RL algorithm RFQI also shows comparable performance under some perturbations. In this paragraph, we want to show that DR-SAC

requires much less training time than RFQI, improving its applicability to large scale problems. Table 1 lists the training time of SAC, DR-SAC, FQI, and RFQI algorithms in three testing environments. DR-SAC is demonstrated to be well-trained in at most $20\%$ time required by RFQI. Compared with each non-robust counterpart, the training time of DR-SAC is at most $3.6\times$ of SAC, while RFQI requires $10-13\times$ more training time than FQI. In Figure 7, we provide a plot of performance changes against the training time in the *Reacher* environment, where RFQI is shown to be under-trained when the curve of DR-SAC converges.

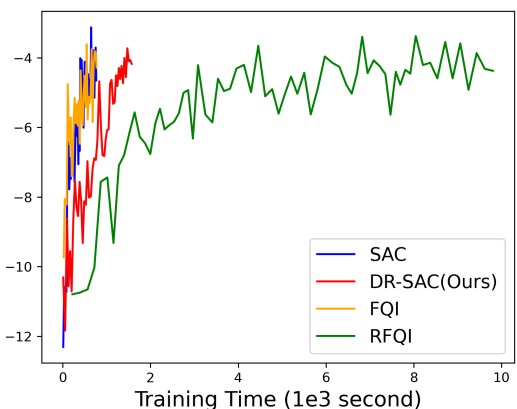

Figure 7: Average Reward of 20 Episodes over Training Time in *Reacher* Environment.

Moreover, this efficiency improvement does not solely arise from the functional approximation step, but also from the inherent optimization efficiency in the loss function structure. The RFQI algorithm considers the RMDP framework with uncertainty sets defined by the TV distance and is empirically built on the BCQ algorithm. In RFQI, there exists a step similar to Equation (22) to find the optimal functional under empirical measurement. Experimental results show that the efficiency gap arises from the number of GD steps in solving this optimization problem. RFQI sets the default GD steps as 1000 while DR-SAC achieves comparable robustness performance with only 5 steps. To further investigate, we vary the GD steps in RFQI to 5, 10 and 100 in the *LunarLander* environment and report the model performance in the unperturbed environment. As shown in Table 6, performance drops sharply when RFQI uses fewer GD steps, indicating that the loss function structure in RFQI inherently leads to slower convergence and requires more optimization steps. In our framework, the choice of actor-critic based non-robust baseline, KL divergence induced uncertainty set and generative modeling in nominal distribution estimation together yields a more optimization-friendly formulation, contributing to our method's practical efficiency.

Table 6: GD steps, training time and performance in *LunarLander*

| Algorithm | DR-SAC | RFQI | RFQI | RFQI | RFQI (Used) |
|---|---|---|---|---|---|
| GD Steps | 5 | 5 | 10 | 100 | 1000 |
| Training Time (min) | 36 | 12 | 21 | 139 | 238 |
| Nominal Env Performance | 240.0 | 175.9 | 181.9 | 192.9 | 201.2 |

### C.3.2 SELECTION OF GENERATIVE MODEL

**Robustness of VAE.** A consistent challenge in DR-RL algorithm design is that unknown nominal distributions $p_{s,a}^0$ often appear in the loss function. In Section 3.2 and Appendix A.1, we review methods used in other model-free DR-RL algorithms and motivate the necessity of generative models in our setting. Although generative models inevitable introduce additional estimation error when constructing empirical measures $\tilde{p}_{s,a}^0$, our ablation studies demonstrate that DR-SAC is largely insensitive to the VAE modeling, therefore improving its applicability. In the *Pendulum* environment, where the state and action space dimensions are 3 and 1 respectively, we train DR-SAC with VAEs of latent dimensions $1, 5, 10, 20, 50$ and evaluate performance under perturbed pendulum mass. As shown in Figure 3(a), DR-SAC maintains superior robustness over the SAC baseline as long as the latent dimension lies within a reasonable range (between 5 and 20 in our experiments).

**Comparison with Other Generative Models.** To demonstrate the choice of VAE over other generative models, we implemented Diffusion Probabilistic Models and Normalizing Flows as alternatives to the VAE in DR-SAC and conducted ablation studies on *Pendulum* and *Cartpole*. Model performance is provided in Figure 3(b), (c) and Figure 8. Flow-based models showed unstable performance even in unperturbed *Pendulum* environment. DR-SAC with Diffusion models achieved comparable robustness to the VAE in *Pendulum*. Crucially, the efficiency of sampling process with Diffusion models is a major bottleneck. Diffusion-based training is at least $4.5\times$ slower than VAE-based training.

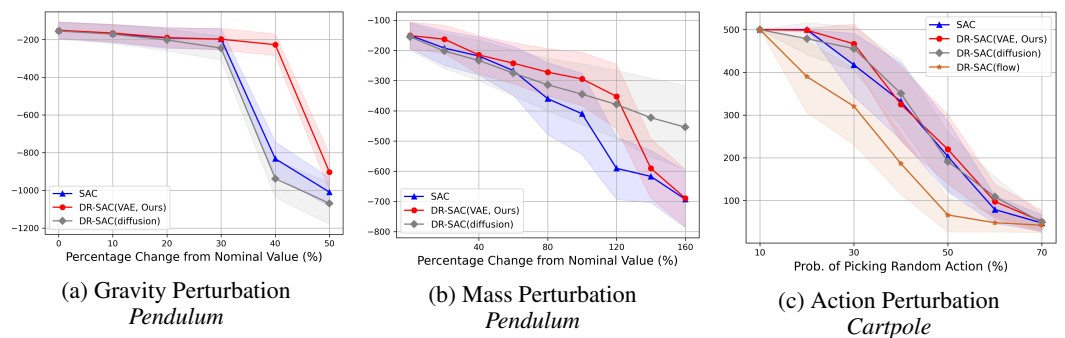

(a) Gravity Perturbation
*Pendulum*

(b) Mass Perturbation
*Pendulum*

(c) Action Perturbation
*Cartpole*

Figure 8: Additional results with different generative models in *Pendulum* and *Cartpole*.

### C.3.3 USAGE OF V-NETWORK

In this section, we demonstrate that keeping the $V$-network in the SAC algorithm reduces the sensitivity on dataset distribution. As introduced in Appendix C.1, offline datasets in this work are generated by first training a behavior policy and applying the epsilon-greedy method to collect data. Experimental results shows that SAC without the $V$-network exhibits unstable performance when the behavior policy differs across datasets.

Our experiments are conducted in the *Pendulum* environment. We generate two datasets with behavior policy trained by an online version of SAC and TD3, denoted as SAC-dataset and TD3-dataset, respectively. Figure 9 presents the average reward of 20 episodes against training steps in four scenarios: SAC-dataset vs. TD3-dataset, SAC algorithm with vs. without $V$-network. Removing the $V$-network shows minor influence on offline SAC learning using SAC-dataset. However, for TD3-dataset, SAC with $V$-network achieves a stable average reward around $-150$ quickly, but the average reward of SAC without $V$-network fluctuates intensely and never exceeds $-200$. This validates that SAC with a $V$-network is less sensitive to behavior policy and dataset distribution.

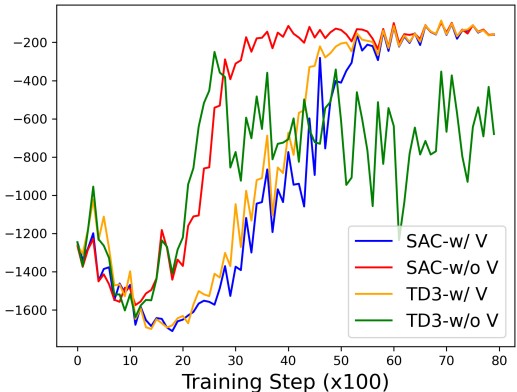

Figure 9: Average Reward of 20 Episodes over Training Step in *Pendulum* Environment.

## D  REGRET BOUND

**Definition D.1.** The distributionally robust regret $R_{\mathcal{M}_\delta}(\pi)$ of a policy $\pi \in \Pi$ is defined as:
$$R_{\mathcal{M}_\delta}(\pi) := \left\| V_{\mathcal{M}_\delta}^\star - V_{\mathcal{M}_\delta}^\pi \right\|_\infty.$$

For any policy $\pi$, the soft value and soft $Q$-functions satisfy:
$$V_{\mathcal{M}_\delta}^\pi(s) = \mathbb{E}_{a \sim \pi(\cdot|s)} \left[ Q_{\mathcal{M}_\delta}^\pi(s,a) - \alpha \log \pi(a \mid s) \right].$$
The following inequality holds:
$$\left\| V_{\mathcal{M}_\delta}^\star - V_{\mathcal{M}_\delta}^\pi \right\|_\infty \leq \left\| Q_{\mathcal{M}_\delta}^\star - Q_{\mathcal{M}_\delta}^\pi \right\|_\infty.$$

Based on the estimator $\hat{p}_{s,a}^0$, we define the corresponding estimate DR soft value function
$$\hat{V}_{\mathcal{M}_\delta}^\pi(s) = \inf_{\mathbf{p} \in \hat{\mathcal{P}}(\delta)} \mathbb{E}_\mathbf{p} \left[ \sum_{t=1}^\infty \gamma^{t-1} \left( r_t + \alpha \cdot \mathcal{H}\left(\pi\left(s_t\right)\right) \right) \mid \pi, s_1 = s \right],$$
where $\hat{\mathcal{P}}_{s,a}(\delta) := \left\{ p_{s,a} \in \Delta(|\mathcal{S}|) : D_{\mathrm{KL}}\left(p_{s,a} \| \hat{p}_{s,a}^0\right) \leq \delta \right\}$. Similarly, the estimate DR soft $Q$-function is given by
$$\hat{Q}_{\mathcal{M}_\delta}^\pi(s,a) = \inf_{\mathbf{p} \in \hat{\mathcal{P}}(\delta)} \mathbb{E}_\mathbf{p} \left[ r_1 + \sum_{t=2}^\infty \gamma^{t-1} \left( r_t + \alpha \cdot \mathcal{H}\left(\pi\left(s_t\right)\right) \right) \mid \pi, s_1 = s, a_1 = a \right].$$

Define $\hat{V}_{\mathcal{M}_\delta}^\star = \max_{\pi \in \Pi} \hat{V}_{\mathcal{M}_\delta}^\pi$ and $\hat{\pi}_{\mathcal{M}_\delta}^\star \in \operatorname{argmax}_{\pi \in \Pi} \hat{V}_{\mathcal{M}_\delta}^\pi$.

The estimate $\hat{\mathcal{T}}_\delta^\pi$ is defined as
$$\hat{\mathcal{T}}_\delta^\pi Q(s,a) = \mathbb{E}[r] + \gamma \cdot \sup_{\beta \geq 0} \left\{ -\beta \log \left( \mathbb{E}_{\hat{p}_{s,a}^0} \left[ \exp\left( \frac{-V(s')}{\beta} \right) \right] \right) - \beta\delta \right\},$$

**Assumption D.2.** Assume $\mathrm{KL}(p_{s,a}^0 \| \hat{p}_{s,a}^0) \leq \varepsilon_1^2$ and $\mathrm{supp}(p_{s,a}^0) = \mathrm{supp}(\hat{p}_{s,a}^0)$.

By Pinsker's inequality, $\mathrm{TV}(p_{s,a}^0, \hat{p}_{s,a}^0) \leq \frac{1}{2}\sqrt{\mathrm{KL}(p_{s,a}^0 \| \hat{p}_{s,a}^0)} \leq \frac{1}{2}\varepsilon_1$.

**Bound of $\|\hat{\mathcal{T}}_\delta^\pi Q(s,a) - \mathcal{T}_\delta^\pi Q(s,a)\|$.**
**Lemma D.3.** *Under Assumption D.2,*
$$\left\| \hat{\mathcal{T}}_\delta^\pi Q(s,a) - \mathcal{T}_\delta^\pi Q(s,a) \right\| \leq 2\gamma\varepsilon_1 \frac{R_{\max} + \alpha \log |A|}{(1-\gamma)\delta} e^{(R_{\max} + \alpha \log |A|)/(1-\gamma)\underline{\beta}}.$$

*Proof.* As we defined in Section 3.2,
$$f((s,a),\beta) := -\beta \log\left( \mathbb{E}_{p_{s,a}^0}\left[ e^{-V(s')/\beta} \right] \right) - \beta\delta, \quad \hat{f}((s,a),\beta) := -\beta \log\left( \mathbb{E}_{\hat{p}_{s,a}^0}\left[ e^{-V(s')/\beta} \right] \right) - \beta\delta.$$

From (Xu 2023, Proposition 5), the maximums of $f((s,a),\beta)$ and $\hat{f}((s,a),\beta)$ are achieved at $\beta^\star, \hat{\beta}^\star \in [0, V_{\max}/\delta]$, that is,
$$\mathcal{T}_\delta^\pi Q(s,a) = \mathbb{E}[r] + \gamma \sup_{\beta \geq 0} f((s,a),\beta) = \mathbb{E}[r] + \gamma \sup_{\beta \in [0, V_{\max}/\delta]} f((s,a),\beta),$$
$$\hat{\mathcal{T}}_\delta^\pi Q(s,a) = \mathbb{E}[r] + \gamma \sup_{\beta \geq 0} \hat{f}((s,a),\beta) = \mathbb{E}[r] + \gamma \sup_{\beta \in [0, V_{\max}/\delta]} \hat{f}((s,a),\beta).$$
Hence,
$$\left| \hat{\mathcal{T}}_\delta^\pi Q(s,a) - \mathcal{T}_\delta^\pi Q(s,a) \right| \leq \gamma \sup_{\beta \in [0, V_{\max}/\delta]} \left| \hat{f}((s,a),\beta) - f((s,a),\beta) \right|.$$

Note that $\mathrm{supp}(p_{s,a}^0) = \mathrm{supp}(\hat{p}_{s,a}^0)$, which implies that $F_p(0) = \operatorname{essinf}_{s' \sim p_{s,a}^0} V(s') = \operatorname{essinf}_{s' \sim \hat{p}_{s,a}^0} V(s') = F_{\hat{p}}(0)$. Now, we can assume that the optimal $\beta^\star, \hat{\beta}^\star$ is achieved in $[\underline{\beta}, V_{\max}/\delta]$, where $\underline{\beta} = \min\{\beta^\star/2, \hat{\beta}^\star/2, 1/2\}$.

Then, we aim to bound the supremum of $\left|\hat{f}((s,a),\beta) - f((s,a),\beta)\right|$ over the interval $[\underline{\beta}, V_{\max}/\delta]$. Since $\log x \le x - 1$ when $x \ge 1$, we have

$$|\hat{f}((s,a),\beta) - f((s,a),\beta)| = \beta \left| \log\left(\mathbb{E}_{\hat{p}^0_{s,a}}\left[e^{-V(s')/\beta}\right]\right) - \log\left(\mathbb{E}_{p^0_{s,a}}\left[e^{-V(s')/\beta}\right]\right)\right|$$

$$\le \frac{V_{\max}}{\delta} \frac{\left|\mathbb{E}_{\hat{p}^0_{s,a}}\left[e^{-V(s')/\beta}\right] - \mathbb{E}_{p^0_{s,a}}\left[e^{-V(s')/\beta}\right]\right|}{\min\left\{\mathbb{E}_{\hat{p}^0_{s,a}}\left[e^{-V(s')/\beta}\right], \mathbb{E}_{p^0_{s,a}}\left[e^{-V(s')/\beta}\right]\right\}}.$$

Since $\mathrm{TV}(p^0_{s,a}, \hat{p}^0_{s,a}) \le \varepsilon_1$ and $V_{\min} \ge 0$, we have

$$\left|\mathbb{E}_{\hat{p}^0_{s,a}}\left[e^{-V(s')/\beta}\right] - \mathbb{E}_{p^0_{s,a}}\left[e^{-V(s')/\beta}\right]\right| = \left|\sum_{s'\in S}(\hat{p}^0_{s,a}(s') - p^0_{s,a}(s'))e^{-V(s')/\beta}\right|$$

$$\le \sum_{s'\in S}\left|\hat{p}^0_{s,a}(s') - p^0_{s,a}(s')\right|e^{-V_{\min}/\beta} \le 2\mathrm{TV}(p^0_{s,a}, \hat{p}^0_{s,a}) \le \varepsilon_1.$$

In addition,

$$\min\left\{\mathbb{E}_{\hat{p}^0_{s,a}}\left[e^{-V(s')/\beta}\right], \mathbb{E}_{p^0_{s,a}}\left[e^{-V(s')/\beta}\right]\right\} \ge e^{-V_{\max}/\underline{\beta}}.$$

Thus, we obtain

$$|\hat{f}((s,a),\beta) - f((s,a),\beta)| \le \varepsilon_1 \frac{V_{\max}}{\delta} e^{V_{\max}/\underline{\beta}}, \quad \text{where } V_{\max} = \frac{R_{\max} + \alpha\log|A|}{1-\gamma}.$$

Combining this with the earlier inequality gives

$$\left\|\hat{\mathcal{T}}^\pi_\delta Q(s,a) - \mathcal{T}^\pi_\delta Q(s,a)\right\| \le \gamma\varepsilon_1 \frac{R_{\max} + \alpha\log|A|}{(1-\gamma)\delta} e^{(R_{\max}+\alpha\log|A|)/(1-\gamma)\underline{\beta}} := \varepsilon_2.$$

$\square$

**Bound of** $\|\hat{Q}^\pi_{\mathcal{M}_\delta} - Q^\pi_{\mathcal{M}_\delta}\|$. For any $\pi \in \Pi$, let $Q^{k+1} = \mathcal{T}^\pi_\delta Q^k$, $\hat{Q}^{k+1} = \hat{\mathcal{T}}^\pi_\delta \hat{Q}^k$, and $\hat{Q}^0 = Q^0$. By Proposition 3.4, we know that $Q^k$ will converge to the DR soft Q-value $Q^\pi_{\mathcal{M}_\delta}$, which is the fixed point of $T^\pi_\delta$. That is, $T^\pi_\delta Q^\pi_{\mathcal{M}_\delta} = Q^\pi_{\mathcal{M}_\delta}$ and $Q^k \to Q^\pi_{\mathcal{M}_\delta}$. Similarly, there exists a fixed point of $\hat{T}^\pi_\delta$ such that $\hat{T}^\pi_\delta \hat{Q}^\pi_{\mathcal{M}_\delta} = \hat{Q}^\pi_{\mathcal{M}_\delta}$ and $\hat{Q}^k \to \hat{Q}^\pi_{\mathcal{M}_\delta}$. Then

$$\|\hat{Q}^\pi_{\mathcal{M}_\delta} - Q^\pi_{\mathcal{M}_\delta}\| = \|\hat{T}^\pi_\delta \hat{Q}^\pi_{\mathcal{M}_\delta} - T^\pi_\delta Q^\pi_{\mathcal{M}_\delta}\|$$

$$= \|\hat{T}^\pi_\delta \hat{Q}^\pi_{\mathcal{M}_\delta} - T^\pi_\delta \hat{Q}^\pi_{\mathcal{M}_\delta} + T^\pi_\delta \hat{Q}^\pi_{\mathcal{M}_\delta} - T^\pi_\delta Q^\pi_{\mathcal{M}_\delta}\|$$

$$\le \varepsilon_2 + \gamma\|\hat{Q}^\pi_{\mathcal{M}_\delta} - Q^\pi_{\mathcal{M}_\delta}\|$$

$$\implies \|\hat{Q}^\pi_{\mathcal{M}_\delta} - Q^\pi_{\mathcal{M}_\delta}\| \le \frac{\varepsilon_2}{1-\gamma}.$$

**Regret bound.** We define the updating policy as

$$\hat{\pi}_{k+1} = \operatorname*{argmin}_{\pi\in\Pi} D_{\mathrm{KL}}\left(\pi(\cdot\mid s) \left\| \frac{\exp\left(\frac{1}{\alpha}\hat{Q}^{\hat{\pi}_k}_{\mathcal{M}_\delta}(s,\cdot)\right)}{Z^{\hat{\pi}_k}(s)}\right.\right), k = 0, 1, \cdots$$

By Proposition 3.6, the policy sequence $\{\hat{\pi}^k\}$ converges to the optimal policy $\hat{\pi}^\star_{\mathcal{M}_\delta}$ under the estimate DR soft policy iteration as $k \to \infty$.

For each state $s \in \mathcal{S}$, we have $V^\star_{\mathcal{M}_\delta}(s) - V^{\hat{\pi}^\star_{\mathcal{M}_\delta}}_{\mathcal{M}_\delta}(s) \ge 0$. By definition, $\hat{V}^\star_{\mathcal{M}_\delta}(s) = \hat{V}^{\hat{\pi}^\star_{\mathcal{M}_\delta}}_{\mathcal{M}_\delta}(s)$. Then, we have

$$V^\star_{\mathcal{M}_\delta}(s) - V^{\hat{\pi}^\star_{\mathcal{M}_\delta}}_{\mathcal{M}_\delta}(s) \le \left|V^\star_{\mathcal{M}_\delta}(s) - \hat{V}^\star_{\mathcal{M}_\delta}(s)\right| + \left|\hat{V}^{\hat{\pi}^\star_{\mathcal{M}_\delta}}_{\mathcal{M}_\delta}(s) - V^{\hat{\pi}^\star_{\mathcal{M}_\delta}}_{\mathcal{M}_\delta}(s)\right|$$

$$= \left|\sup_\pi V^\pi_{\mathcal{M}_\delta}(s) - \sup_\pi \hat{V}^\pi_{\mathcal{M}_\delta}(s)\right| + \left|\hat{V}^{\hat{\pi}^\star_{\mathcal{M}_\delta}}_{\mathcal{M}_\delta}(s) - V^{\hat{\pi}^\star_{\mathcal{M}_\delta}}_{\mathcal{M}_\delta}(s)\right|$$

$$\le \sup_\pi \left|V^\pi_{\mathcal{M}_\delta}(s) - \hat{V}^\pi_{\mathcal{M}_\delta}(s)\right| + \left|\hat{V}^{\hat{\pi}^\star_{\mathcal{M}_\delta}}_{\mathcal{M}_\delta}(s) - V^{\hat{\pi}^\star_{\mathcal{M}_\delta}}_{\mathcal{M}_\delta}(s)\right|$$

$$\le 2\sup_\pi \left|V^\pi_{\mathcal{M}_\delta}(s) - \hat{V}^\pi_{\mathcal{M}_\delta}(s)\right|$$

Thus,

$$R_{\mathcal{M}_\delta}(\hat{\pi}^\star_{\mathcal{M}_\delta}) = \left\| V^\star_{\mathcal{M}_\delta} - V^{\hat{\pi}^\star_{\mathcal{M}_\delta}}_{\mathcal{M}_\delta} \right\|_\infty \leq 2 \sup_\pi \left\| V^\pi_{\mathcal{M}_\delta} - \hat{V}^\pi_{\mathcal{M}_\delta} \right\|_\infty \leq 2 \sup_\pi \left\| Q^\pi_{\mathcal{M}_\delta} - \hat{Q}^\pi_{\mathcal{M}_\delta} \right\|_\infty \leq \frac{2\varepsilon_2}{1 - \gamma}.$$

