# OpenReview forum: "DR-SAC: Distributionally Robust Soft Actor-Critic for Reinforcement Learning under Uncertainty"
_ICLR.cc/2026/Conference — ICLR 2026 Poster_

### Official Review · Reviewer_aZrL · 2025-10-22

**Soundness:** 3
**Presentation:** 3
**Contribution:** 3
**Rating:** 6
**Confidence:** 3

**Summary:**

This paper proposes Distributionally Robust Soft Actor-Critic (DR-SAC), a model-free reinforcement learning algorithm that integrates distributionally robust optimization into the Soft Actor-Critic framework. The key idea is to handle environment uncertainty via a KL-divergence–bounded uncertainty set over transition distributions. The authors derive a distributionally robust soft Bellman operator using KL duality, which introduces a scalar dual variable $\beta$ for each $(s,a)$ pair. To avoid solving numerous local optimizations, they replace these with a single functional optimization $g(s,a)$ over the entire dataset, justified by interchange theorems from variational analysis. To handle continuous state–action spaces and offline settings, the paper incorporates a VAE-based generative model to approximate the nominal transition distribution $p^0(s'|s,a)$, thereby mitigating the double-sampling issue that arises from nonlinear expectations in the KL dual form.

**Strengths:**

- **Methodological innovation**: Introducing KL-based DR soft Bellman operator and replacing numerous scalar $\beta$-optimizations with a single functional optimization are novel and computationally effective.
- **Efficiency-performance trade-off**: Demonstrates superior robustness compared to SAC/RFQI while reducing training time.

**Weaknesses:**

1. **Theory–implementation gap**: The convergence and monotonicity proofs apply only to tabular finite-action cases, whereas the implemented DR-SAC relies on continuous Gaussian policies and neural approximators.
2. **VAE dependency**: While the need for a generative model (to avoid double-sampling) is well justified, empirical validation of the VAE’s adequacy is limited (mostly latent-dimension sweeps) and lacks comparisons with alternative generative models such as flows or diffusion models.
3. **Limited robustness evaluation**: Perturbations focus on Gaussian noise and parameter scaling. Tests under more challenging conditions (e.g., heavy-tailed, regime-switching, or reset/restart scenarios) are absent. The adequacy of the KL-ball assumption versus, e.g., Wasserstein metrics remains unexamined.

**Questions:**

1. With errors from function approximation, VAE modeling, and functional optimization present, can the authors provide partial error–convergence guarantees (e.g., one-step contraction with bias)?
2. What is the quantitative rationale for choosing a VAE over flow-, diffusion-, or score-based transition models?
3. Do the authors believe the KL-ball assumption adequately captures realistic structural or regime shifts? Would a Wasserstein-ball or f-divergence family extension offer advantages?
4. (minor) Please fix typographical errors ("In iour algorithm") and ensure consistent terminology (e.g., "KL ball").

---

> ### Author Response · Authors · 2025-11-23
> **Response to Reviewer aZrL (Part 1/2)**
>
> **Summary of Response:**
> We thank Reviewer aZrL for the insightful comments and for recognizing the methodological innovation and efficiency of our DR-SAC algorithm. In this response, we address the key concerns by:
>
> 1.  **Comparing Generative Models:** We provide new ablation studies comparing VAE against **Diffusion** and **Normalizing Flow** models. Results show that VAE strikes the best balance—achieving robustness comparable to Diffusion models but with **4-5x faster training speed**, making it the most scalable choice.
> 2.  **Addressing Theoretical Gaps:** We clarify that the continuous-action gap is a shared limitation with standard SAC and emphasize our empirical contribution.
> 3.  **Expanding Robustness Tests:** We added experiments with **Heavy-tailed (Cauchy) noise**, demonstrating DR-SAC's resilience to non-Gaussian perturbations.
> 4.  **Justifying KL-Divergence:** We elaborate on the computational intractability of Wasserstein distances in high-dimensional continuous spaces to justify our KL-based design choice.
>
>
> **Detailed Response:**
>
> **1. Response to Weakness 1: Theory-Implementation Gap**
>
> We acknowledge this theoretical gap. As the reviewer correctly identifies, this limitation originates from our baseline, **SAC**, which also assumes finite action spaces in its rigorous analysis (Haarnoja et al., 2018) while using function approximators in practice. The primary challenge is that the entropy-augmented value function can become unbounded in continuous spaces without restrictive assumptions.
> **Our Position:** Our contribution is to extend this empirically successful framework to the robust setting. We demonstrate the distributionally robust optimization technique effectively bridges this gap in practice, providing significant robustness improvements. Formalizing convergence for maximum entropy RL in continuous spaces remains an open problem in the broader field.
>
> **2. Response to Weakness 2 & Question 2: VAE vs. Diffusion/Flow Models**
>
> We appreciate this suggestion. To validate our choice, we implemented **Diffusion Probabilistic Models** and **Normalizing Flows** as alternatives to the VAE in DR-SAC and conducted ablation studies on *Pendulum* and *Cartpole*.
>
> **Results (details in Figure 9 and 10 in Appendix C.3.2):**
>
>   * **Performance:** DR-SAC with Diffusion models achieved robustness comparable to (or slightly better than) the VAE in *Pendulum*. However, Flow-based models showed unstable performance.
>   * **Efficiency (Key Rationale):** Crucially, the **sampling speed** of Diffusion models become a major bottleneck. As shown in the table below, Diffusion-based training is at least **4.5x slower** than VAE-based training.
>   * **Conclusion:** We chose VAE because it offers the best **efficiency-performance trade-off**, enabling our algorithm to scale to larger problems where Diffusion models would be computationally prohibitive.
>
> *Table: Training Time (Normalized to SAC) & Performance under Perturbation*
> | Env | Model | Training Time | Performance (High Perturbation) |
> | :--- | :--- | :--- | :--- |
> | **Pendulum** | **VAE (Ours)** | **\~2.5x SAC** | **-352.2** (Robust) |
> | | Diffusion | \~11.5x SAC | -378.9 (Robust) |
> | **Cartpole** | **VAE (Ours)** | **\~2.0x SAC** | **Success** |
> | | Diffusion | \~5.5x SAC | Success |
> | | Flow | \~2.5x SAC | Failed |
>
> *(Note: Specific values are updated in the revised Appendix.)*
>
> **3. Response to Weakness 3: Limited Robustness Evaluation**
>
>
>   * **Heavy-tailed Noise:** We conducted a new experiment on *Pendulum* adding **Cauchy-distributed noise** (undefined variance, heavy tails) to state observations.
>       * **Result:** DR-SAC significantly outperforms baselines. For example, at noise scale 0.075, DR-SAC achieves reward **-474.5** vs. SAC's **-561.7** and FQI's **-809.0**. (See Figure 2(c) in revised version).
>   * **Regime Switching:** Our framework assumes an $(s,a)$-rectangular uncertainty set (independent uncertainty per state-action pair). Systematic structural shifts like global regime switching or resets violate this independence assumption and typically require different mathematical frameworks (e.g., POMDPs or Contextual MDPs). We focus on transition uncertainty, which covers a broad class of physical perturbations (mass, friction, noise).

---

> ### Author Response · Authors · 2025-11-23
> **Response to Reviewer aZrL (Part 2/2)**
>
> **4. Response to Question 1: Partial Error-Convergence Guarantees**
>
> Establishing a rigorous one-step contraction bound with function approximation, VAE error, and functional optimization error is extremely challenging and constitutes a separate theoretical research topic (similar to the open problems in analyzing SAC itself).
> However, intuitively:
>
> 1.  **VAE Consistency:** Under well-covered data observations, existing literature confirms that VAEs are consistent estimators of the distribution density [1,2]
>
> 2. **Partial Convergence result under VAE Error**  In the revised version Appendix D, we have added a theoretical result showing how the VAE transition-model error influences the one-step contraction in DR soft-Q iteration, and consequently the final robust value gap relative to the optimal robust policy. We hope this provides a better theoretical understanding of how VAE error affects the convergence of policy learning.
>
> 3.  **Functional Approximation:** Our Proposition 3.7 shows the functional optimization is equivalent to the scalar optimization in the limit.
>     Thus, assuming the VAE error and Neural Network approximation errors are bounded by $\epsilon$, the error in our Bellman operator application is bounded. We leave the formal derivation of this compound error bound for future work.
>
> **5. Response to Question 3: KL vs. Wasserstein**
>
> While Wasserstein distance has geometric advantages, we chose KL-divergence for **computational tractability**, which is critical for scalability:
>
> 1.  **Closed-Form Dual:** KL-divergence allows us to derive an elegant strong dual (Proposition 3.3) involving a simple scalar optimization ($\sup_{\beta \ge 0}$).
> 2.  **Scalability:** This scalar dual form enables our **Functional Optimization** trick (Proposition 3.7), reducing the problem to a single optimization pass.
> 3.  **Curse of Dimensionality:** The dual form of Wasserstein distance typically involves an inner minimization over the entire state space ($\inf_{s'' \in \mathcal{S}} \{V(s'') + \lambda d(s', s'')\}$). In continuous high-dimensional spaces, solving this inner problem is computationally prohibitive (often requiring discretization). KL-divergence avoids this entirely.
>
> We have corrected all typos mentioned. Thank you for helping us strengthen our paper.
>
>
> **Refs:**
>
> [1] Tang, Rong, and Yun Yang. "On empirical bayes variational autoencoder: An excess risk bound." Conference on Learning Theory. PMLR, 2021.
>
> [2] Surendran, Sobihan, Antoine Godichon-Baggioni, and Sylvain Le Corff. "Theoretical convergence guarantees for variational autoencoders." arXiv preprint arXiv:2410.16750 (2024).

---

> ### Comment · Reviewer_aZrL · 2025-11-23
>
> Thank you for the thorough and well-organized rebuttal.
> The additional experiments and clarifications effectively address my main concerns.
> I have revised my rating accordingly.

---

> > ### Author Response · Authors · 2025-11-23
> >
> > Dear Reviewer aZrL,
> >
> > Thank you very much for your positive feedback and for reconsidering your rating.
> >
> > We strictly believe that your insightful comments were extremely valuable and have significantly contributed to making our work more solid and rigorous. We strictly appreciate the time and effort you dedicated to reviewing our paper!
> >
> > Best regards, The Authors

---

### Official Review · Reviewer_ku4q · 2025-10-31

**Soundness:** 2
**Presentation:** 1
**Contribution:** 2
**Rating:** 4
**Confidence:** 4

**Summary:**

This work proposes a DR-SAC algorithm under a KL-divergence-based uncertainty set. It employs a Variational Autoencoder (VAE) to estimate the empirical distribution from the offline dataset. The proposed algorithm is model-free and can be applied to environments with continuous action spaces.

**Strengths:**

1. This algorithm can be applied in continuous action/state space setting.

2. Using VAE to avoid double sampling in offline DR-RL problem.

3. This work provide convergence guarantee.

**Weaknesses:**

1. The presentation of this work is not very clear or professional. The gradients used in the algorithm section are not defined where they first appear—I had to consult the appendix to understand their meaning. Moreover, the assumptions on which the convergence analysis relies are not stated explicitly, making it difficult to follow the theoretical results.

2. The proof section lacks sufficient rigor. The main theorem is presented without clearly stating the required assumptions, and the accompanying proof is incomplete and lacks detail. A more thorough and transparent exposition of the assumptions and proof steps is necessary to make the theoretical claims convincing.

**Questions:**

1. Could the authors provide a more detailed discussion on why the VAE can effectively avoid the double-sampling process, and whether the estimated distribution satisfies the requirement of being an unbiased estimator? In previous offline DR-RL works, the use of double-sampling and treating the empirical distribution derived from offline data has been theoretically justified — how does this approach maintain similar rigor?

2. Are there any additional requirements or assumptions regarding the offline dataset (e.g., data coverage, distributional support, or sample quality) for the proposed method to perform effectively?

---

> ### Author Response · Authors · 2025-11-23
> **Response to Reviewer ku4q (Part 1/2)**
>
> **Summary of Response:**
> We sincerely thank Reviewer ku4q for their rigorous review and for identifying key areas where the presentation and theoretical clarity could be improved. We have taken your feedback very seriously. In this revision, we have:
>
> 1.  **Rewritten Section 3 and the Appendix** to explicitly state all gradient definitions and theoretical assumptions (specifically $|\mathcal{A}| < \infty$ for bounded entropy), ensuring the proofs are rigorous and self-contained.
> 2.  **Clarified the VAE's role:** We explain that while the VAE estimator is biased (consistent), it is a necessary trade-off to enable KL-based robustness in continuous spaces where standard empirical risk minimization fails (degenerates to non-robust solutions).
> 3.  **Justified the Offline Dataset Requirements:** We clarify that our method shares the same data coverage assumptions as standard offline RL (e.g., SAC), supported by empirical evidence.
>
> **1. Response to Weakness 1: Presentation and Gradient Definitions**
>
> We apologize for the oversight regarding the gradient definitions. We agree that clear notation is crucial for readability.
>
>   * **Revision:** We have updated Algorithm 1 and the main text to explicitly define the empirical gradient estimates for $\psi$, $\phi$, and $\alpha$. These follow the standard SAC formulation:
>       * $\hat{\nabla}\_{\psi}J\_{V}(\psi)$ targets the squared residual of the value function.
>       * $\hat{\nabla}\_{\phi}J\_{\pi}(\phi)$ minimizes the KL-divergence between the policy and the Boltzmann distribution.
>       * $\hat{\nabla}\_{\alpha}J(\alpha)$ adjusts the entropy temperature.
>
> **2. Response to Weakness 2: Theoretical Rigor and Assumptions**
>
> We appreciate this feedback and have revised Section 3.1 to explicitly state the assumptions required for each theoretical result. The core assumption is **bounded entropy**, which typically requires a finite action space assumption ($|\mathcal{A}| < \infty$) in theoretical analysis, a standard practice in entropy-regularized RL literature [1].
>
> To be precise, we have clarified the assumptions for each proposition:
>
>   * **Proposition 3.3 (Dual Operator):** Requires bounded $Q(s,a)$ values to ensure the existence of the moment generating function.
>   * **Proposition 3.4 (Policy Evaluation):** Holds for a fixed policy with a fixed entropy term; explicit $|\mathcal{A}| < \infty$ is not strictly required if $Q^0$ is bounded.
>   * **Proposition 3.5 (Policy Improvement):** Requires $|\mathcal{A}| < \infty$ to guarantee the existence of an optimal policy maximizing the KL-divergence of new policy and the exponential Q-function.
>   * **Theorem 3.6 (Convergence):** Relies on the above propositions and thus assumes $|\mathcal{A}| < \infty$ for theoretical rigor. In our continuous implementation, this is practically handled by clipping the standard deviation of the Gaussian policy to prevent unbounded entropy.
>
> **3. Response to Question 1: VAE and Double Sampling**
>
> **Why VAE avoids double sampling:**
> The double-sampling issue in KL-constrained Robust RL arises because the dual objective involves a non-linear term: $\log(\mathbb{E}_{s' \sim P^0}[\exp(\dots)])$.
>
>   * **Failure of naive ERM:** If we use the single sample $s'$ from the offline dataset to estimate the inner expectation, the term simplifies to $\log(\exp(\dots))$, which cancels out the non-linearity. As derived in our Appendix A.1 (and noted in your comment), this causes the robust Bellman operator to **degenerate** into the standard, non-robust Bellman operator ($\mathcal{T}_{\delta, g^*}^\pi Q \approx \mathcal{T}^\pi Q$).
>   * **VAE Solution:** The VAE learns the conditional density $P(s'|s,a)$. This allows us to **generate multiple synthetic samples** $\tilde{s}'\_1, \dots, \tilde{s}'\_k$ for a single $(s,a)$ pair. We use these generated samples to construct a Monte-Carlo estimate of the inner expectation $\mathbb{E}\_{s' \sim \hat{P}\_{VAE}}[\dots]$, preserving the non-linear structure of the robust objective.
>
> **Bias and Consistency:**
>
>   * **Unbiasedness:** You are correct that the VAE-based estimator is **not unbiased**. Generally, density estimation in model-based RL introduces bias.
>   * **Consistency:** However, standard results in generative modeling (e.g., [2,3]) suggest that VAEs are **consistent** estimators. As the number of data points $N \to \infty$, the estimated distribution $\hat{P} \to P^0$. We show in Appendix D that if the VAE error is bounded by $\epsilon$, the error in the robust Bellman operator is also bounded. Thus, while biased in finite samples, it is a consistent approach that enables tractability in continuous spaces.

---

> ### Author Response · Authors · 2025-11-23
> **Response to Reviewer ku4q (Part 2/2)**
>
> **4. Response to Question 2: Justification in Previous Works**
>
> Previous works justified estimation from offline data by restricting the problem setting, which does not apply to our general continuous case:
>
> 1.  **Linear Settings:** Works like [Panaganti et al., 2022] use **Total Variation (TV)** distance. The dual form of TV is linear in the inner expectation, so the inner and outer expectations commute, removing the double-sampling issue entirely.
> 2.  **Discrete States:** Works like [Shi et al., 2024] rely on revisiting exact $(s,a)$ pairs to compute empirical frequencies, which is impossible in continuous spaces.
> 3.  **R-Contamination:** Works like [Wang et al., 2021] use a specific uncertainty set (mixture of nominal and arbitrary) that simplifies the objective.
>     **Our Rigor:** We address the general case with continuous space and a KL-divergence uncertainty set. Since we cannot use the algebraic tricks from TV or discrete frequencies, we rely on the **generative consistency** of the VAE. This is a necessary trade-off: we accept model approximation error (consistency instead of unbiasedness) to gain the expressiveness of KL-robustness in continuous control.
>
> **5. Response to Question 3: Dataset Assumptions**
>
> Our method does not impose *additional* theoretical assumptions beyond standard offline RL algorithms like SAC.
>
>   * **Data Coverage:** Like SAC, DR-SAC requires the dataset to have sufficient coverage of the state-action space to learn a valid Q-function. If the dataset is too sparse, both the Q-function and the VAE transition model will generalize poorly. In the revised version, we have added a theoretical result on how the VAE transition-model error influences the final robust value gap relative to the optimal robust policy. We hope this provides a better theoretical understanding of what happens when the VAE cannot consistently estimate the model (and, on the other hand, it also gives a guarantee of convergence to the optimal policy when the VAE is consistent).
>
>   * **Empirical Observation:** In our experiments (e.g., Appendix C.3.3), we found that DR-SAC works effectively on datasets where standard SAC works. We strictly followed the standard D4RL/offline RL benchmarks (datasets generated by behavior policies with epsilon-greedy exploration) to ensure fair comparisons.
>
>
> We hope these clarifications and revisions address your concerns regarding rigor and presentation. Thank you again for improving our paper.
>
>
> **Ref:**
>
> [1] Haarnoja, Tuomas, et al. "Soft actor-critic: Off-policy maximum entropy deep reinforcement learning with a stochastic actor." International conference on machine learning. Pmlr, 2018.
>
> [2] Tang, Rong, and Yun Yang. "On empirical bayes variational autoencoder: An excess risk bound." Conference on Learning Theory. PMLR, 2021.
>
> [3] Surendran, Sobihan, Antoine Godichon-Baggioni, and Sylvain Le Corff. "Theoretical convergence guarantees for variational autoencoders." arXiv preprint arXiv:2410.16750 (2024).

---

> ### Comment · Reviewer_ku4q · 2025-11-28
>
> Thanks for the authors' response. My concerns have been addressed, and I will raise my score.

---

### Official Review · Reviewer_B6wb · 2025-11-01

**Soundness:** 2
**Presentation:** 3
**Contribution:** 3
**Rating:** 6
**Confidence:** 3

**Summary:**

This paper tackles the problem of developing a distributionally robust version of the Soft Actor-Critic (SAC) for offline reinforcement learning (RL) from a dataset with a continuous action space. Specifically, the authors develop a maximum entropy framework for policy iteration to accomplish the RL objective across an uncertainty set of transitions defined by KL-divergence, showing that their algorithm converges. They introduce a technique that reformulates the harder state-action optimizations into a single optimization problem over a function space. Through the use of a Variational Autoencoder (VAE), they are able to estimate the nominal transitions and thus generate samples without requiring access to a simulator.

**Strengths:**

1. The paper is well written and easy to follow. Namely, the paper systematically addresses the challenges discussed in section 3.1 in the subsequent sections.
2. The use of interchanging minimization and integration in a decomposable space under the stated assumption is clever.
3. The introduction of a VAE to estimate the nominal transition kernel for a given state-action pair, $p_{s,a}^0$ alleviates the double-sampling issue encountered in the calculation of the empirical risk by producing synthetic samples. This combined with the above point and the use of a neural network allows for the extension to a continuous action space.
4. Extensive experimentation indicates the validity of the author's theoretical claims.

**Weaknesses:**

1. There appears to be a fundamental disconnect between the convergence guarantees in Theorem 3.6 (which rely on the assumption that $|\mathcal{A|<\infty}$ and hence tabular) and the practical extension to continuous actions in the algorithm's construction. Namely, the convergence guarantees are not formally proven under continuous actions, but rather implied through the use of a neural network and empirical validation.
2. The use of a VAE to approximate the nominal transition kernel introduces additional risk due to the optimization occurring on this estimation and not the true environment. Should a given dataset not have sufficient coverage, the VAE would learn a model that is not necessarily robust to the true environmental shifts, but rather be robust to some other environmental shifts. The authors should consider including further discussion on this topic in the ablation studies in section 4.3 or additional theory-based discussion on this in the appendix. Can you provide a reference that provides some additional clarity on the above point?

**Questions:**

1. On line 122, adding "discounted" in front of "Markov Decision Process" adds clarity to exactly what problem setting you are considering.
2. Extra closing parenthesis on line 125.
3. Inconsistent notation of $\Delta(\mathcal{S}), \Delta(S),$ and $\Delta(|\mathcal{S}|)$ on line 126 and equation 5.
4. Addition of $\forall s\in\mathcal{S}$ to make equations 6-8 more precise.
5. Use of $|A|<\infty$ in Assumption 3.1 versus $|\mathcal{A}|<\infty$ elsewhere.
6. Please clarify what values $\tau$ takes, and it's impact in the algorithm.
7. It would help to make your work more appealing to a wider audience unfamiliar with deep learning by precisely clarifying that $\hat{\nabla}$ denotes the empirical stochastic gradient estimate in Algorithm 1.
8. Is the standard deviation plotted in figure 1.e?
9. Typo of "grind" instead of "grid" on lines 1190 and 1240.
10. Typo of "larges" on line 1379.
11. Addition of grid lines in figures 7 and 9 for consistency.

---

> ### Author Response · Authors · 2025-11-23
> **Response to Reviewer B6wb (Part 1/2)**
>
> **Summary of Response:**
> We appreciate Reviewer B6wb's positive assessment of our paper's clarity, the novelty of our functional optimization technique, and the effectiveness of the VAE integration. In this response, we strictly address the two main concerns: **1)** We acknowledge the theoretical gap regarding continuous action spaces, clarifying that this is a shared limitation with the standard SAC baseline, and emphasize our contribution lies in empirically extending distributional robustness to this challenging setting. **2)** We discuss the implications of VAE approximation errors, citing our ablation studies to demonstrate that DR-SAC remains robust even with varying VAE capacities, and comparing its data requirements to standard model-free methods. Finally, we have corrected all identified typos and clarified the notation and hyperparameters as suggested.
>
> **1. Response to Weakness 1: Theoretical Guarantees vs. Continuous Implementation**
>
> We acknowledge this important theoretical gap. As the reviewer correctly identifies, this limitation originates from our baseline algorithm, SAC, which also assumes finite action spaces in the theoretical analysis [1,2] while using neural network function approximators in practice. The primary theoretical difficulty lies in the fact that the entropy-augmented value function can become unbounded in continuous action spaces without restrictive assumptions. To the best of our knowledge, establishing formal convergence guarantees for entropy-regularized RL in continuous spaces remains an open problem in the broader RL community.
>
> **Our Position:** Our contribution is to extend the empirically successful SAC framework to a **distributionally robust (DR)** setting. We demonstrate that by incorporating the distributionally robust optimization technique, we can achieve superior robustness against extensive types of perturbations. We agree that this is a shared limitation with the current entropy-regularized RL literature. In our revised paper, we will explicitly state this distinction: while our convergence proofs hold for the tabular case (providing a theoretical foundation), the continuous control implementation is an empirical extension validated by extensive experiments.
>
> **2. Response to Weakness 2: VAE Estimation and Data Coverage**
>
> This is a very valid concern for any algorithm involving model estimation in offline RL.
>
>   * **Performance robustness to VAE misspecification:** We agree that if the dataset lacks coverage in certain state-action regions, the VAE may learn an inaccurate nominal model. However, we respectfully argue that this dependency is not unique to our VAE-based approach; standard model-free offline RL (like SAC or CQL) also requires sufficient coverage to learn accurate Q-functions. Theoretically, we have added a new perturbation analysis in Appendix D to illustrate how the bias incurred by the VAE error, under the TV norm, propagates from the true transition model to the value gap between the policy learned by DR-SAC and the optimal robust value under the true transition. We hope this additional result provides theoretical justification for our VAE-based approach.
>
>
>   * **Robustness to VAE Quality:** Empirically, our algorithm shows resilience to VAE modeling choices. As detailed in **Appendix C.3.2**, we conducted ablation studies varying the VAE's latent dimension. Results show that DR-SAC maintains consistent performance and robustness improvements across a wide range of hyperparameter choices, suggesting that "perfect" model estimation is not a strict prerequisite for the benefits of distributional robustness to materialize.
>   * **Revision:** Following your suggestion, we will add a dedicated discussion in Section 4.3 comparing the data coverage requirements of model-based components vs. pure Q-learning, and explicit caveats about model bias in sparse-data regimes.
>
> **Ref:**
>
> [1] Haarnoja, Tuomas, et al. "Soft actor-critic: Off-policy maximum entropy deep reinforcement learning with a stochastic actor." International conference on machine learning. Pmlr, 2018.
>
> [2] Haarnoja, Tuomas, et al. "Soft actor-critic algorithms and applications." arXiv preprint arXiv:1812.05905 (2018).

---

> ### Author Response · Authors · 2025-11-23
> **Response to Reviewer B6wb (Part 2/2)**
>
> **3. Response to Questions & Minor Issues**
>
>   * **Q1-Q5 (Clarity & Notation):** We thank the reviewer for the meticulous reading. We have fixed the "discounted" phrasing, the extra parentheses, the inconsistent notation of $\Delta\mathcal{S}$, and the missing of $\forall s \in \mathcal{S}$ in Equations 6-8. We have also standardized the use of $|\mathcal{A}|$ across the paper.
>   * **Q6 (Value of $\tau$):** $\tau$ represents the soft update coefficient for the target networks (Polyak averaging). It is fixed at **0.005** in all our experiments to ensure training stability, consistent with the standard SAC implementation.
>   * **Q7 (Gradient Notation):** We will clarify in the text that $\hat{\nabla}$ denotes the empirical stochastic gradient estimate, consistent with standard deep learning literature.
>   * **Q8 (Figure 1.e Standard Deviation):** In Figure 1(e), the standard deviation regions were computed but omitted from the final plot because the overlapping shaded areas of multiple algorithms made the figure unreadable. We have added Table 4 in Appendix C.2 to present the standard deviation clearly.
>   * **Typos:** We have corrected "grind" to "grid" and "larges" to "large" as pointed out.
>
> We hope these revisions address the reviewer's concerns, and thank you again for helping improve the quality of our manuscript.

---

### Official Review · Reviewer_ftXw · 2025-11-01

**Soundness:** 3
**Presentation:** 3
**Contribution:** 3
**Rating:** 6
**Confidence:** 3

**Summary:**

The paper introduce a new  Distributionally Robust SAC algorithm. Theoretical analysis and numerical examples are given to show the advantages of the proposed methods.

**Strengths:**

The paper introduce a new  Distributionally Robust SAC algorithm. Theoretical analysis and numerical examples are given to show the advantages of the proposed methods.

**Weaknesses:**

The benchmarks used are low dimension examples. It would be good to add some higher dimension examples with continuous action spaces such as Ant in Mujoco environments.

**Questions:**

Is there any limitation for the general distributional robust RL framework?
Is it possible to learn the uncertainty of the environment and onine adapt?

---

> ### Author Response · Authors · 2025-11-23
>
> **Summary**
>
> We thank Reviewer ftXw for recognizing our work's theoretical analysis and empirical advantages. In this response, we have  **clarified the distinction between offline DR-RL and online adaptation**, discussing the trade-off between conservatism and safety in real-world deployments where online retraining is risky or infeasible.
>
>
> **Detailed Response:**
>
> **1. Response to Weakness: High-dimensional Benchmarks**
>
> We thank the reviewer for this constructive suggestion. We fully recognize the value of validating our method on higher-dimensional tasks like Ant. We have indeed conducted preliminary investigations on the Ant environment. However, we observed that standard **offline** baselines (e.g., CQL) rarely report consistent performance or robustness metrics on Ant in the literature [1, 2], likely due to the task's high instability and sensitivity to hyperparameter tuning in offline settings. The lack of established "robustness baselines" on Ant makes it difficult to perform a rigorous and fair comparative analysis.
>
> Our Focus on HalfCheetah with Uncertainty: Consequently, we prioritized HalfCheetah (State dim: 17, Action dim: 6), which is widely accepted as a representative high-dimensional benchmark in the Distributionally Robust RL (DR-RL) community. To ensure the evaluation was challenging and comprehensive, we went beyond standard setups by introducing explicit perturbations (e.g., varying torso mass and friction coefficients) identical to the uncertainty sets defined in our theory.
> As shown in Figure 1, our DR-SAC significantly outperforms standard SAC in maintaining stability and survival rates under these high-dimensional perturbations, effectively demonstrating the scalability of our robust framework.
>
>
> **2. Response to Question: Limitations of DR-RL and Online Adaptation**
>
> This is an insightful question that touches on the fundamental design philosophy of Robust RL versus Adaptive RL.
>
> **On the Limitations of DR-RL:**
> You are correct that the primary limitation of the general DR-RL framework is potential **conservatism**. By optimizing for the worst-case scenario within an uncertainty set (as defined in our Eq. 5), the algorithm slightly sacrifices performance in the nominal environment to ensure robustness across a range of perturbed environments.
>
> **On Online Adaptation vs. Offline Robustness:**
> Regarding the possibility of learning uncertainty and adapting online, we offer the following perspective:
>
> 1.  **Different Problem Settings:** Our work focuses on the **Offline RL** setting, where the agent must learn a robust policy solely from a static dataset without interacting with the environment. Online adaptation implies trial-and-error in the real world. In many safety-critical real-world applications (e.g., medical treatment, industrial robotics), such errors can be dangerous or prohibitively expensive.
> 2.  **The Role of DR-RL and Possible Direction for Online Adaptation:** The goal of DR-SAC is to produce a policy that is "robust by design", capable of handling environmental shifts *immediately* upon deployment without needing further training or adaptation, which matches the nature of offline RL. Extending such an approach to online settings is indeed a promising direction with increasing attention. Recent works like [3-5] explore online DR-RL, and others like [6] focus on hybrid or transfer learning approaches to handle shifts. While online adaptation can reduce conservatism, it requires a mechanism to safely update the model in real-time, which remains a distinct challenge from the offline robustness problem we address here. We believe exploring the intersection of these fields is a fruitful direction for future work.
>
>
> Thank you again for helping us clarify the scope and contribution of our work.
>
>
>
> **Ref:**
>
> [1] Haarnoja, Tuomas, et al. "Soft actor-critic algorithms and applications." arXiv preprint arXiv:1812.05905 (2018).
>
> [2] Kumar, Aviral, et al. "Conservative q-learning for offline reinforcement learning." Advances in neural information processing systems 33 (2020): 1179-1191.
>
> [3] Lu, Miao, et al. "Distributionally robust reinforcement learning with interactive data collection: Fundamental hardness and near-optimal algorithms." Advances in Neural Information Processing Systems 37 (2024): 12528-12580.
>
> [4] Ghosh, Debamita, George K. Atia, and Yue Wang. "Provably near-optimal distributionally robust reinforcement learning in online settings." arXiv preprint arXiv:2508.03768 (2025).
>
> [5] Liu, Zhishuai, and Pan Xu. "Distributionally robust off-dynamics reinforcement learning: Provable efficiency with linear function approximation." International Conference on Artificial Intelligence and Statistics. PMLR, 2024.
>
> [6] Qu, Chengrui, et al. "Hybrid transfer reinforcement learning: Provable sample efficiency from shifted-dynamics data." arXiv preprint arXiv:2411.03810 (2024)

---

> > ### Comment · Reviewer_ftXw · 2025-11-26
> >
> > Thanks for the authors' clarifications. I have raised my score.

---

> > > ### Author Response · Authors · 2025-11-26
> > >
> > > Dear Reviewer ftXw,
> > >
> > > Thank you very much for your positive feedback and for reconsidering your rating.
> > >
> > > We strictly believe that your insightful comments on the limitations of DR-RL were extremely valuable and have significantly contributed to making our work more solid and rigorous. We strictly appreciate the time and effort you dedicated to reviewing our paper!
> > >
> > > Best regards, The Authors

---

### Author Response · Authors · 2025-12-02
**Main Concerns and Our Responses**

# Overall
We summarize main concerns by topic and present how our responses addressed these concerns. There are now **no remaining unresolved correctness issues**. All remaining points are clearly stated limitations shared with standard SAC or offline RL, rather than flaws of our approach. Given the strengthened theoretical discussion, new experiments, and updated reviewer evaluations, we hope this summary assists the AC in making a confident decision.

## 1. Theory–Implementation Gap

**Concern:** Reviewer aZrL noted that our convergence results are proved in tabular settings, while the implemented DR-SAC uses continuous actions and neural networks.

**Response:** We clarified that this is a **shared limitation with standard SAC**: existing SAC convergence analyses assume finite action spaces and bounded entropy, whereas practical implementations use continuous Gaussian policies with neural network function approximation. Our theoretical contribution is to establish **DR soft policy iteration** in the same tabular regime as SAC, and our empirical contribution is to **extend this framework to a robust continuous-control implementation**. We do not break any theoretical result that SAC has.

**Feedback:** Reviewer aZrL stated that concern was addressed and raised score.

---
## 2. Generative Model Design and Double-Sampling

**Concerns:**
- Reviewer ku4q requested a more detailed explanation of how the VAE avoids double-sampling, especially compared to prior offline DR-RL works that rely on empirical distributions.
- Reviewer B6wb was concerned about VAE quality under limited data coverage.
- Reviewer aZrL asked for comparisons with alternative generative models.

**Response and New Experiments:**
- We provided a detailed explanation of **how double-sampling arises** in the general KL-robust setting and why naive empirical risk minimization causes the robust Bellman operator to **degenerate into the non-robust operator**.
- We clarified that the VAE-based estimator is **biased but consistent**. Its quality depends on sufficient data coverage, but this dependence is **standard in offline RL** (e.g., SAC/CQL also require sufficient coverage). In the revised version, we added:
  - A **theoretical bound** on the error of the DR soft Bellman operator when the VAE error is bounded.
  - Experiments showing that DR-SAC remains robust under **different VAE latent dimensions**.
- We implemented **Diffusion Probabilistic Models** and **Normalizing Flows** as replacements for the VAE. Results show that diffusion models achieve similar robustness but are **4–5× slower**, while flow-based models exhibit unstable performance. In general, VAEs provide the **best robustness–efficiency trade-off**.

**Feedback:** Reviewers ku4q and aZrL indicated that clarifications and new experiments resolved their concerns and raised their scores.

---
## 3. Choice of KL-Divergence

**Concern:** Reviewer aZrL questioned the choice of a KL-ball uncertainty set over Wasserstein or general f-divergences.

**Response and Justification:**
We clarified that KL-divergence was chosen for **computational tractability and scalability**:
- KL admits a **closed-form dual**, yielding a single scalar dual variable per state–action pair and enabling our **functional optimization trick**.
- Wasserstein duals require **inner minimization over the state space**, which is computationally prohibitive in high-dimensional continuous RL settings and would render our algorithm impractical.

We do not claim KL is universally superior; we now present it explicitly as a **pragmatic and scalable design choice**, and highlight Wasserstein/f-divergence extensions as promising future work.

**Feedback:** Reviewer aZrL confirmed this clarification addressed the concern.

---
## 4. Experimental Scope and Perturbation Types

**Concern:** Reviewer ftXw requested experiments on higher-dimensional environments such as Ant, and Reviewer aZrL asked for more challenging perturbations beyond Gaussian noise and basic parameter scaling.

**Response and New Experiments:**
- We clarified that standard offline baselines **rarely report stable or reproducible metrics on Ant** due to its pronounced instability and sensitivity to data coverage in the offline setting, making robust comparison difficult. Instead, we use **HalfCheetah** (state dim 17, action dim 6) with structured perturbations (mass, friction) aligned with our theoretical uncertainty sets, which is a widely used and validated high-dimensional benchmark in DR-RL.
- We added new experiments with **Cauchy-distributed (heavy-tailed) observation noise**, showing that DR-SAC significantly outperforms SAC and RFQI under such perturbations.
- We discussed that perturbations like **global regime switching** violate rectangular uncertainty assumptions and are more appropriately handled with POMDP frameworks.

**Feedback:** Reviewer ftXw and aZrL confirmed that clarifications resolved their concerns and raised their scores.

---

### Author Response · Authors · 2025-12-02
**Summary of Reviewer Feedback and Rebuttal Actions**

We thank the reviewers and the area chair for their careful reading and constructive feedback.

In the **initial reviews**, the reviewers highlighted the following strengths of the proposed Distributionally Robust Soft Actor-Critic (DR-SAC):

- The DR-SAC framework brings **KL-based distributional robustness** to **offline RL with continuous action spaces**, filling an important gap in DR-RL.

- Replacing a collection of scalar dual optimization problems with a **single functional optimization** substantially improves computational efficiency with only minor robustness loss.

- **Using VAEs** to approximate the nominal transition kernel **avoids the double-sampling issue** in the KL-robust dual formulation.

- Extensive **empirical results** demonstrate improved robustness and a favorable **efficiency–performance trade-off** compared to SAC and other DR-RL baselines.

During the **rebuttal**, we added theory, experiments, and clarifications that addressed the main concerns of **three reviewers, who explicitly stated that their concerns were resolved and raised their ratings accordingly (ftXw, ku4q, aZrL)**. The remaining reviewer (B6wb) was already positive and maintained a rating above the threshold. In summary, we:

- Clarified that the limitation of convergence guarantees to tabular settings originates from **standard SAC theory**, and that our contribution is to extend this **empirically successful framework to the robust setting**, without breaking any theoretical result that SAC has.

- Provided a more rigorous discussion and new results on the **VAE’s consistency, error propagation, and data coverage assumptions**, including a bound on how model error affects the robust Bellman operator and value gap.

- Added new experiments **comparing VAE vs. diffusion/flow generative models** and testing robustness under **heavy-tailed (Cauchy) noise**, thereby strengthening both the methodological justification and robustness evaluation.

- Discussed the choice of **KL-divergence** by highlighting that its closed-form dual enables the functional optimization trick and **practical scalability**, whereas Wasserstein-type duals are computationally prohibitive in our setting.

We provide a summary of main concerns and our responses in another comment. All main concerns have been addressed and there are now **no remaining unresolved correctness issues**. We thank the reviewers and the area chair again for their precious time and constructive feedback.

---

### Meta-Review · Area_Chair_unsQ · 2026-01-07

**Summary:**

This paper proposes a novel actor-critic based algorithm for robust reinforcement learning, and provide a convergence analysis. Reviewers  ftXw, B6wb, and aZrL provide positive assessments, and the primary concerns raised by reviewer ku4q have been properly addressed in the rebuttal. Overall, review comments acknowledge that this paper is well-written, and the proposed algorithm enjoys advantages of being applicable to continuous action/state space setting and avoiding double sampling. Based on these assessments, I agree with reviewers and recommend acceptance.

**Reviewer Concerns:**

Most reviewer concerns have heen addressed by the rebuttal, such as concerns regarding assumptions of convergence analysis, and clarify of generative model design.

**Reviewer Scores:**

Reviewers ftXw, ku4q, and aZrL stated that they are willing to increase their scores.

---

### Decision · Program_Chairs · 2026-01-26

Accept (Poster)